# High-energy resolution X-ray spectroscopy reveals bonding characteristics of La³⁺ homologues of actinium radiopharmaceuticals
Harry Ramanantoanina [1] ✉, Bianca Schacherl [1] ✉, Attila Kovács[2], Michelangelo Tagliavini[3], Emily Marie Reynolds [1], Cedric Reitz [1], Ruwini S. K. Ekanayake[1], Martin Schäfer[4], Paul-Valentin von Massow[1], Jörg Göttlicher[5], Ralph Steininger[5], Kathy Dardenne [1], Maurits W. Haverkort [3] ✉, Martina Benešová-Schäfer[6] ✉ & Tonya Vitova [1] ✉

Spectroscopic techniques are essential for accurately probing the electronic structures of coordination compounds and revealing the nature of their chemical bonding. This is particularly relevant for *f*-elements, where bonding interactions play crucial role, particularly in radiopharmaceutical developments. We present advanced spectroscopic analysis, including core-to-core resonant inelastic X-ray scattering (CC-RIXS) and high-energy resolution X-ray absorption near-edge structure (HR-XANES), to investigate metal-ligand interactions using lanthanum (La) as a non-radioactive homologue of actinium (Ac) applied in emerging and highly potent therapeutic radiopharmaceuticals. By analyzing the interplay between La 4*f* and 5*d* orbitals in various environments, we extract key information about ligand-field effects and bond covalency. Our findings demonstrate that spectroscopic features of the La $L_2$-edge CC-RIXS map reflect the nephelauxetic effect, which arises from central-field 4*f* orbital-specific bond covalency. The energy separation between the pre-edge and main absorption edge of the La $L_2$-edge HR-XANES spectra also serves as direct probe of electron density for both 4*f* and 5*d* orbitals. Quantum chemical modeling, including ligand-field density-functional theory (LFDFT) and ab initio bond analysis, complements our experiments. This allows us to establish a direct correlation between spectroscopic observables and theoretical metrics for bonding properties, offering a framework to understand the coordination chemistry of *f*-elements. Beyond advancing fundamental chemistry, our findings will also inform future studies on Ac³⁺-radiopharmaceutical agents, where precise knowledge of bonding interactions is essential for their development.

Actinium (Ac) and its isotopes have unique properties that make them interesting, particularly in the field of nuclear medicine[1–3]. Ac-225, a radioisotope with 10 days half-life, has been explored for its applications in targeted alpha therapy[4–10]. In this form of therapy, alpha particle-emitting radionuclides are used to selectively target and destroy cancer cells with minimal damage to surrounding healthy organs and tissue[11,12]. The coordination chemistry of the Ac³⁺ ion plays important role here, as it can influence the overall stability, targeting, pharmacokinetics, and biodistribution, thereby the treatment efficacy and side effects, of ²²⁵Ac-based radiopharmaceuticals[2,13].

---

[1]Institute for Nuclear Waste Disposal (INE), Karlsruhe Institute of Technology (KIT), Karlsruhe, Germany. [2]Joint Research Centre (JRC), European Commission, Karlsruhe, Germany. [3]Institute for Theoretical Physics, University of Heidelberg, Heidelberg, Germany. [4]Service Unit for Radiopharmaceuticals and Preclinical Trials, German Cancer Research Center (DKFZ), Heidelberg, Germany. [5]Institute for Photon Science and Synchrotron Radiation (IPS), Karlsruhe Institute of Technology (KIT), Karlsruhe, Germany. [6]Research Group Translational Radiotheranostics, German Cancer Research Center (DKFZ), Heidelberg, Germany. ✉e-mail: harry.ramanantoanina@kit.edu; bianca.schacherl@kit.edu; m.w.haverkort@thphys.uni-heidelberg.de; m.benesova@dkfz-heidelberg.de; tonya.vitova@kit.edu

Therefore, much attention has been paid to optimize the selectivity and effectiveness of the radiopharmaceuticals[14]. Much less is known about the bonding properties and stability of the complexes under real physiological conditions, making this area of research critical for advancing long-sought cancer treatment strategies. Besides, practical investigations of these radiopharmaceuticals are still hindered by the limited availability of $^{225}$Ac sources, the small amounts of radioisotope obtainable, and the general need for specialized infrastructure[15,16]. Because of this, current chemical research is often based on the study of the lanthanide homologue[17–19] and, to some extent, theoretical modeling[20–22].

However, many questions regarding the binding properties and reactivity remain open, and can only be addressed through advanced spectroscopic methods combined with quantum chemical calculations. Lanthanides and actinides are both *f*-elements, sharing the common feature of valence electrons occupying high-angular momentum *f*-orbitals ($l = 3$): 4*f* and 5*f*, respectively. The actinide 5*f* orbitals have more pronounced interactions (bond covalency) with their chemical environment, being more diffuse than the more tightly bound lanthanide 4*f* orbitals, distinguishing actinides behavior from that of lanthanides[23–25]. Consequently, although direct interpolation of the chemical properties from lanthanides to actinides must be approached carefully, analyzing periodic trends across these series can provide valuable insight into bonding interaction. This helps in understanding actinides' chemical behavior and in predicting their reactivity patterns.

In this work, we focus on lanthanum (La) as a homologue of Ac to develop spectroscopic methods for probing the metal–ligand interactions with high sensitivity. Specifically, we employ advanced X-ray spectroscopic method[26–29], core-to-core resonant inelastic X-ray scattering (CC-RIXS) and high-energy resolution X-ray absorption near-edge spectroscopy (HR-XANES) at the La $L_2$-edge, to investigate the role of the valence 4*f* and 5*d* orbitals in the La-ligand bonding interaction. By preparing La$^{3+}$ complexes in solution under pharmacologically relevant conditions[30], we aim at assessing the extent of metal–ligand orbital mixing and ligand–field effects across different ligand environments. We will demonstrate that the pre-edge intensities in the CC-RIXS reflects La 4*f* participation in bonding, while the energy separation between the main absorption edge and the resolved pre-edge peak in the HR-XANES spectra serves as a spectroscopic measure of the overall La-ligand bond covalency. Complementary quantum chemical calculations provide additional insight, establishing a joint experimental/theoretical framework for understanding metal–ligand interactions in *f*-element coordination chemistry. This work also lays the foundation for future studies on actinium complexes.

## Results and discussion
### Structural analysis
Metal chelators play a crucial role in radiopharmaceutical development by securely binding radioactive metal ions and ensuring their stable delivery to target tissues. The three pharmaceutically important coordinating ligands are: *MACROPA*$^{2-}$ ($N, N' - bis[(6 - carboxy - 2 - pyridil)methyl] - 4, 13 - diaza - 18 - crown6$)[31,32], *DOTA*$^{4-}$ ($2, 2', 2'', 2''' - (1, 4, 7, 10 - Tetraazacyclododecane - 1, 4, 7, 10 - tetrayl)tetraacetate$)[33,34], and *PSMA*-617$^{3-}$, a ligand incorporating the *DOTA*$^{4-}$ chelator and already applied in clinical studies[35,36]. *DOTA*$^{4-}$ and *MACROPA*$^{2-}$ are two widely studied chelators with distinct advantages[2]. *DOTA*$^{4-}$, a macrocyclic ligand, is known for its high thermodynamic stability and kinetic inertness, making it suitable for a variety of radiometals, though its slow complexation kinetics can be limiting for some applications[37]. In contrast, *MACROPA*$^{2-}$ offers faster complexation and is particularly well-suited for accommodating larger ions like Ac$^{3+}$, making it a promising candidate for $^{225}$Ac-based targeted alpha therapies[32].

Five complexes are considered: $[La(MACROPA)]^{1+}$, $[La(DOTA)H_2O]^{1-}$, and $[La(PSMA\text{-}617)H_2O]$, as well as $[La(H_2O)_9]^{3+}$ and a complex with *TRIS* (*Tris(hydroxymethyl)aminomethane*) ligand. *TRIS* is used as a cationic organic buffer operating within the pH range 7–9[38]. The molecular complexes are illustrated in Fig. 1a. For the *TRIS* ligand, previous work demonstrated coordination of a single *TRIS* ligand with the La$^{3+}$ ion in the solid state[39]; but coordination of two *TRIS* ligands can also be possible under appropriate conditions[40]. To account for this, we have included single and double *TRIS* coordination in the theoretical study: $[La(TRIS)(H_2O)_6]^{3+}$ and $[La(TRIS)_2(H_2O)_3]^{3+}$. For convenience, unless stated otherwise, abbreviations are used to designate the complexes (Fig. 1a): MACROPA ($[La(MACROPA)]^{1+}$), DOTA ($[La(DOTA)H_2O]^{1-}$), 9H2O ($[La(H_2O)_9]^{3+}$), 1TRIS ($[La(TRIS)(H_2O)_6]^{3+}$), 2TRIS ($[La(TRIS)_2(H_2O)_3]^{3+}$) and PSMA ($[La(PSMA\text{-}617)H_2O]$).

The experimental and theoretical La-ligand bond lengths are shown in Fig. 1b, showing good agreement between the data. The bond lengths are derived from analysis of La $L_3$-edge extended X-ray absorption fine structure (EXAFS) data and optimized geometries of the complexes using density-functional theory (DFT) calculations. The La–O distances in 9H2O vary in a small range: in average 2.54 Å (DFT) and 2.57 Å (EXAFS). As the other extreme, in MACROPA the bond distances range from single bond with the picolinate oxygen (2.44 Å (DFT)) to van der Waals interaction with N of the polycycle. These two weak interactions with the *MACROPA*$^{2-}$ ligand complete the decadentate coordination field around La$^{3+}$, thus it has no free site for coordination of solvent $H_2O$ molecule - the only such case among the present complexes (cf. Fig. 1a).

Another noteworthy feature is the significantly stronger interaction of $La^{3+}$ with the picolinate oxygen in MACROPA (2.44 Å (DFT)) as compared with the aliphatic carboxylic one in DOTA (2.49 Å DFT and 2.53 Å EXAFS), although both donors are part of the carboxylate anions. The latter La–O distances are not much shorter than those with the oxygens in 9H2O, 1TRIS and 2TRIS. The long La–O ($H_2O$) distance in DOTA (2.63 Å (DFT)) and PSMA (2.67 Å (DFT)) reflects the secondary character of the coordinated solvent $H_2O$ to the empty 9$^{th}$ coordination site of La$^{3+}$. In 2TRIS, there

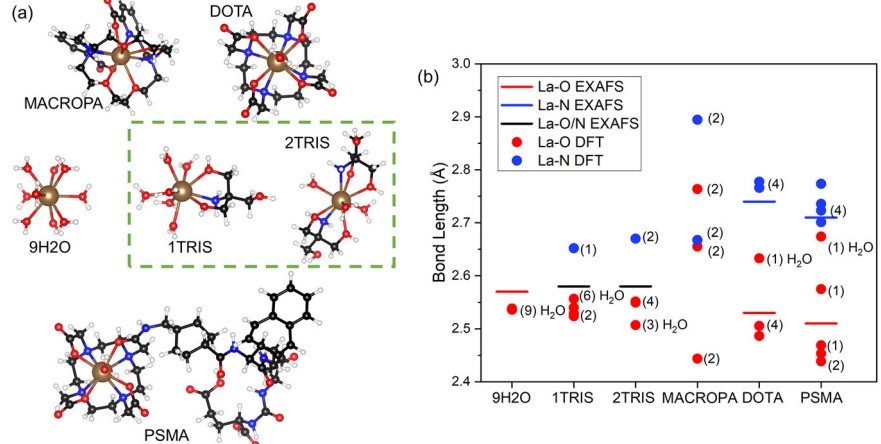

**Fig. 1 | Atomic structures of the La$^{3+}$ homologues for radiopharmaceuticals. a** Ball-and-stick models for $[La(MACROPA)]^{1+}$ (MACROPA), $[La(DOTA)H_2O]^{1-}$ (DOTA), $[La(H_2O)_9]^{3+}$ (9H2O), $[La(TRIS)(H_2O)_6]^{3+}$ (1TRIS), $[La(TRIS)_2(H_2O)_3]^{3+}$ (2TRIS), and $[La(PSMA\text{-}617)H_2O]$ (PSMA). Color code: C (black), H (white), N (blue), O (red), La (brown). **b** Comparison between the experimentally derived (La $L_3$-edge EXAFS) and theoretically calculated (DFT) La–O and La–N bond lengths. Details about the EXAFS analysis are available in the Supplementary Fig. 1 and Supplementary Table 1. The presented theoretical bond lengths correspond to DFT results with the PBE0[92] functional (Supplementary Tables S2–S7). Note that the multiplicity of the La-ligand bonds are also depicted in brackets.

is a marginal increase of the La–N and La–O bonds and an interchange of the relative order of the La–O (*TRIS*) and La–O (*H₂O*) distances with respect to 1TRIS. At a quick glance, the average La-donor distances might suggest similar bonding energies in the six complexes, assuming that the very short La–O distances may compensate for the weaker La–O/N interactions in MACROPA, DOTA and PSMA. We will see in the bond analysis section that the La-ligand distances are misleading in this case.

## La L₂-edge CC-RIXS

The nature of chemical bonding in *f*-element complexes remains one of the most fundamental yet challenging questions in chemistry. The traditional understanding emphasizes the dominant role of the relatively diffuse 5*d* (and 6*s*) orbitals in mediating the metal–ligand interactions, while the localized 4*f* orbitals stay essentially core-like[41–43]. This is with exception to tetravalent systems (examples CeO₂, cerrocene), where spectroscopic and computational studies showed that 4*f* orbitals can display appreciable covalent mixing with ligand orbitals, on the order of several percent[25,44–49]. The contribution of 4*f* orbitals to bonding is generally small in trivalent lanthanide complexes, it is not entirely negligible[25,50] and this might become important, when subtle differences in bonding influence properties. This emphasizes the need to account for both 4*f* and 5*d* orbitals when developing spectroscopic tools. Our spectroscopic approach uses La L₂-edge CC-RIXS, which primarily probes the unoccupied valence 5*d* orbitals of La³⁺ while also providing indirect information on 4*f* (Fig. 2a)[26–29,43,51]. We first establish a theoretical framework to characterize key spectral features, identify the relevant electron transitions, and understand their microscopic origin. We use the ligand–field density-functional theory (LFDFT) model[52] implemented in the Amsterdam modeling suite (AMS)[53] for the calculation.

Figure 2b shows the electronic structures that are involved in CC-RIXS. The ground-state (GS) of the La³⁺ complexes is characterized by a formal 4*f*⁰ configuration resulting with a singlet electronic state derived from the *J* = 0 (¹S) atomic multiplet term. The intermediate-state (IS) involves electronic configurations 2*p*⁵ 4*f*¹ and 2*p*⁵ 5*d*¹, namely the spin-orbit components $2p^1_{1/2} 4f^1$ and $2p^1_{1/2} 5d^1$. The final-state (FS) corresponds to 3*d*⁹ 4*f*¹ and 3*d*⁹ 5*d*¹ configurations. The CC-RIXS process is governed by two main contributions[54]. First, the GS (4*f*⁰) → IS (2*p*⁵ 5*d*¹) → FS (3*d*⁹ 5*d*¹) mechanism is a process with two photons described by two consecutive electric-dipole transitions (*E1*ᵢₙ, *E1*ₒᵤₜ). Second, the GS (4*f*⁰) → IS (2*p*⁵ 4*f*¹) → FS (3*d*⁹ 4*f*¹) mechanism is a process described by electric-quadrupole and electric-dipole transitions (*E2*ᵢₙ, *E1*ₒᵤₜ). The LFDFT method enables the calculation of the multiplet energy levels and ligand–field splitting associated with IS and FS from first principles[52], and the simulation of the corresponding CC-RIXS maps based on the Kramers–Heisenberg formula[55].

Figure 3a shows the calculated CC-RIXS map for La³⁺ free ion, highlighting distinct features that correspond to the main absorption edge (white line, WL) and the pre-edge energy region (PE). The WL is characterized by intense signals driven primarily by the (*E1*ᵢₙ, *E1*ₒᵤₜ) mechanism, while the PE region exhibits weaker features, predominantly originating from the (*E2*ᵢₙ, *E1*ₒᵤₜ) mechanism. Additionally, Fig. 3a displays the calculated atomic multiplet energy levels associated with (2*p*⁵ 4*f*¹, and 2*p*⁵ 5*d*¹) as well as FS (3*d*⁹ 4*f*¹, and 3*d*⁹ 5*d*¹) electronic structures. These multiplet energy levels are aligned with the CC-RIXS map, enabling a direct correlation between individual CC-RIXS features and the underlying electronic structure.

The WL region exhibits a dominant signal centered at an excitation energy of approximately 5915.8 eV, corresponding to a principal peak at a transfer energy of 854.3 eV, accompanied by two satellite features at lower transfer energies. These features are attributed to IS with *J* = 1 (2*p*⁵ 5*d*¹), decaying into final states (FS) with *J* = 0 and *J* = 2, associated with 3*d*⁹ 5*d*¹ multiplet structures. In the PE region, three distinct signals are identified at an excitation energy of around 5905.5 eV. The most intense peak (denoted as peak A in Fig. 3a) appears at a transfer energy of 843.3 eV, while two weaker features (peaks B and C) are observed at 846.1 and 829.3 eV, respectively. Analysis of the multiplet energy levels reveals that these PE features originate from IS with *J* = 2 (2*p*⁵ 4*f*¹), decaying into FS with *J* = 3 and *J* = 1 (3*d*⁹ 4*f*¹). Peaks A and B arise from $3d^3_{3/2} 4f^1$, peak C with higher energy

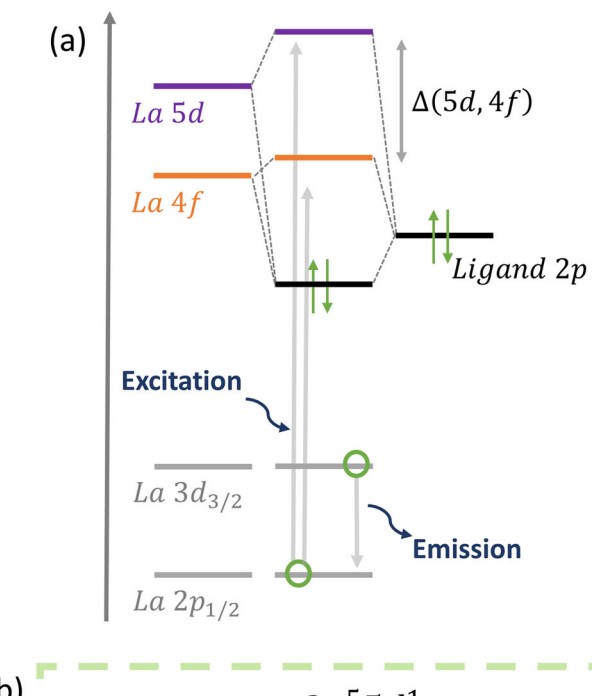

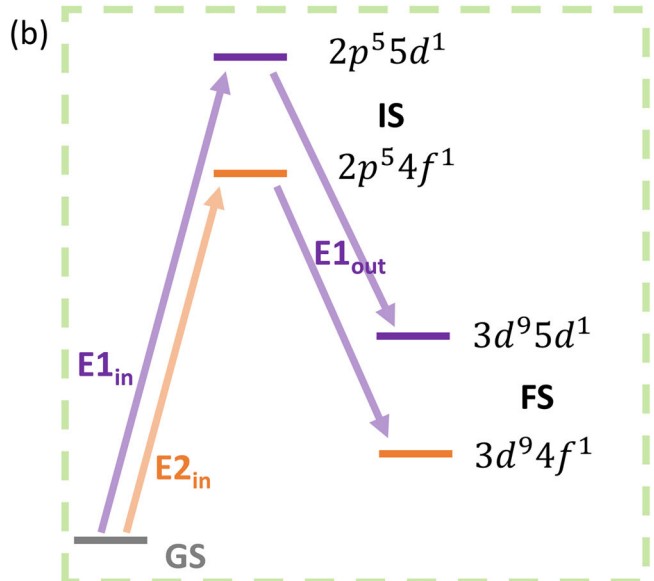

**Fig. 2 | Mechanism of the La L₂-edge CC-RIXS process. a** Representation of two-photon processes whereby X-ray light is absorbed by exciting one electron from the La core 2*p*₁/₂ shell. This is followed by relaxation, where electron in La core 3*d* orbitals fill the 2*p* core hole. **b** Electronic structures that results from the CC-RIXS mechanism: ground-state (GS), excited intermediate-state (IS), and final-state (FS). Note that the electron transitions are governed by the selection rules of both the electric-dipole (*E1*) and electric-quadrupole (*E2*) transitions. The total CC-RIXS signal is given by |(*E1*ᵢₙ + *E2*ᵢₙ)*E1*ₒᵤₜ|² (see also the Supplementary Theoretical Model).

transfer arises from $3d^5_{5/2} 4f^1$, which is reached with weak intensity because of the spin-orbit coupling of the La 3*d* electrons. Thus, the CC-RIXS analysis shows that the three PE signals (A, B, and C) originate from intra-atomic electron-electron interaction in the FS and the spin-orbit coupling of the 3*d* electrons. The calculated branching ratio A:B:C is found to be 0.57:0.41:0.02 for the atomic system.

Figure 3b shows the calculated CC-RIXS map for La³⁺ free ion, presented as a function of the excitation and emission energies. In addition to this map, three selected cross-sections are plotted along the excitation energy axis at constant emission energies, chosen to correspond to the maxima of the WL (black dashed line) and the two prominent PE peaks: A

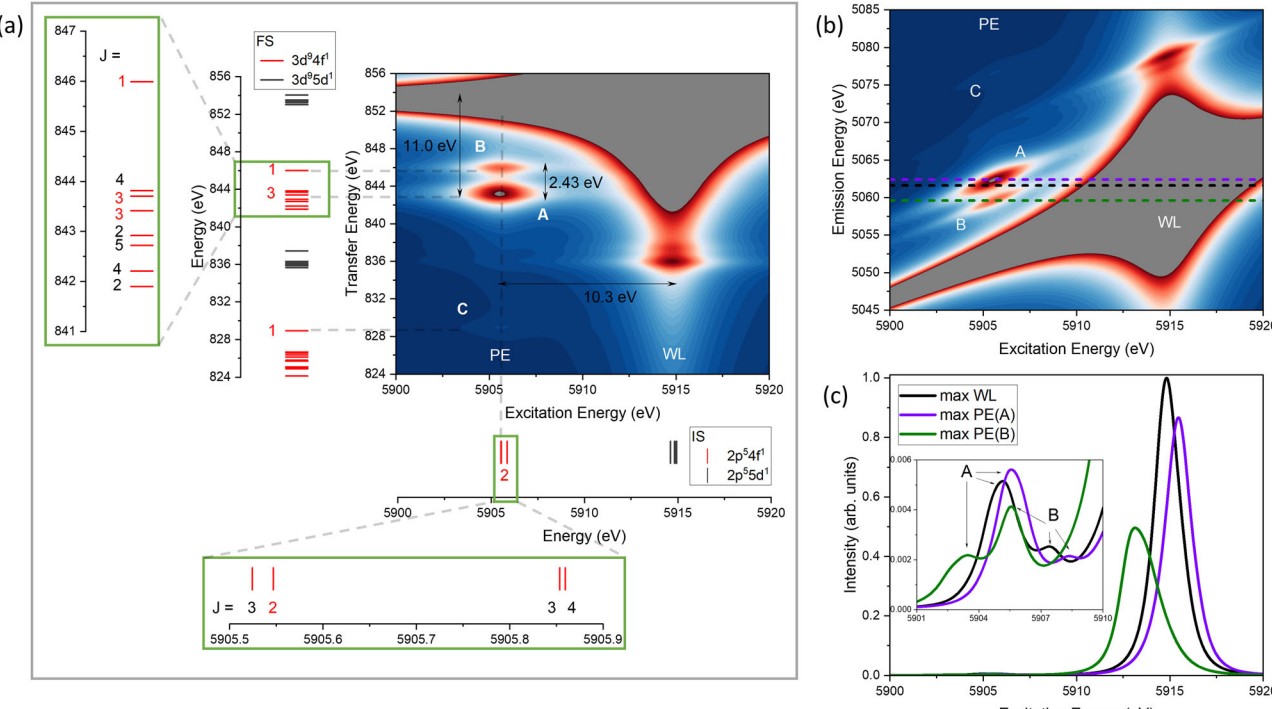

**Fig. 3 | La L$_2$-edge CC-RIXS of La$^{3+}$ free ion. a** Calculated map (LFDFT) showing different features that correspond to the main absorption edge peak (WL) and the pre-edge (PE) A, B, and C signals, together with the calculated multiplet energies of the IS and FS electronic states (red and black bars) of the CC-RIXS process. **b** Translation of the calculated map with respect to the excitation and emission energies, showing also selected cross-sections (colored dashed lines) representing HR-XANES spectra. **c** Calculated HR-XANES spectral profiles for La L$_2$-edge of La$^{3+}$ free ion following selective cross-sections. Note that in **a** the map is represented with respect to the excitation and transfer energies for direct comparison with the electronic structure. The calculations are done using LFDFT with the PBE0[92] DFT functional. Note also that the multiplet energy levels are characterized with their total angular momentum $J$ values. The $2p^5\,4f^1$ (red bars) and $2p^5$

$5d^1$ (black bars) configurations, relevant to IS, give rise to $^{1,3}$D + $^{1,3}$F + $^{1,3}$G and $^{1,3}$P + $^{1,3}$D + $^{1,3}$F atomic multiplet terms (Russell-Saunders coupling). When spin-orbit coupling is introduced, the terms split into total angular momentum $J$ multiplets with $J = 1 + 2 + 2 + 3 + 3 + 4 + 4 + 5$ ($2p^3_{3/2}\,4f^1$), $J = 2 + 3 + 3 + 4$ ($2p^1_{1/2}\,4f^1$), $J = 0 + 1 + 1 + 2 + 2 + 3 + 3 + 4$ ($2p^3_{3/2}\,5d^1$) and $J = 1 + 2 + 2 + 3$ ($2p^1_{1/2}\,5d^1$). The $3d^9\,4f^1$ configuration (red bars), relevant to FS, gives rise to $^{1,3}$P + $^{1,3}$D + $^{1,3}$F + $^{1,3}$G + $^{1,3}$H terms (Russell-Saunders coupling). With spin-orbit coupling, the terms split into $J = {}^1$P$_1$ + $^1$D$_2$ + $^1$F$_3$ + $^1$G$_4$ + $^3$P$_{0,1,2}$ + $^3$D$_{1,2,3}$ + $^3$F$_{2,3,4}$ + $^3$G$_{3,4,5}$ + $^3$H$_{4,5,6}$. Note that in **b** the cross-sections are selectively chosen along the emission energy that corresponds to the maximum of the WL (black dashed line) and PE peaks A (violet dashed line) and B (green dashed line).

(violet dashed line) and B (green dashed line). We observe that WL and the two PE maxima originate from distinct emission energies: i.e., they are not collinear in the CC-RIXS map (Fig. 3b). Specifically, peak A appears at a slightly higher emission energy than the WL, whereas peak B is located at a lower emission energy. Here, the observed non-alignment of WL and PE features can be rationalized based on differences in core-hole screening effects between the $2p$ and $3d$ shells. The calculated map indeed reveals large energy separation ($\Delta E(5d, 4f)$) between the WL and PE of peak A. But $\Delta E(5d, 4f)$ manifests differently along the excitation energy axis (10.3 eV) and the transfer energy axis (11.0 eV) (see Fig. 3a). This induces anisotropic displacement of the spectral features, reflecting distinct relaxation behaviors and electronic structures depending on the core level involved, i.e., slightly weaker core-hole screening effect for IS versus slightly stronger for FS.

Figure 3c presents the calculated HR-XANES spectra derived from selected cross-sections of the CC-RIXS map. The PE region of the HR-XANES spectra is comprised of multiple overlapping features, largely attributable to the convolution of signals arising from the closely spaced peak A and peak B transitions identified in the CC-RIXS map. It is worth noting that PE are generally not observed in conventional XANES measurements[54,56]. Notably, the HR-XANES profile extracted at the WL maximum (black spectra in Fig. 3c), which provides the most direct comparison to experiment, displays a dominant contribution from peak A, with a weaker, low-energy shoulder corresponding to peak B (see inset in Fig. 3c).

To understand the influence of metal–ligand interactions on the structure of the calculated CC-RIXS pre-edge energy region (Fig. 3), we conducted a systematic analysis of the multiplet energy levels by parametrically varying the Slater–Condon integrals. In ligand–field theory[57–59],

the integrals F$^2$($2p$, $4f$), G$^2$($2p$, $4f$), and G$^4$($2p$, $4f$) govern the intra-atomic $2p$-$4f$ electron-electron repulsion in IS, whereas the integrals F$^2$($3d$, $4f$), F$^4$($3d$, $4f$), G$^1$($3d$, $4f$), G$^3$($3d$, $4f$) and G$^5$($3d$, $4f$) govern the core-valence interactions La $3d$-$4f$ in FS. It is well established that ligand coordination leads to a reduction of these integrals relative to the free ion values—a phenomenon known as the nephelauxetic effect[57,60–62]. This effect is typically quantified using the nephelauxetic ratio $\beta$ defined as the scaling factor applied to the free ion Slater–Condon integrals, with $\beta = 1.0$ corresponding to the unperturbed free ion and $\beta \ll 1.0$ indicating to some degree of bond covalency[57,60–64].

Figure 4 shows the FS multiplet energy levels as a function of $\beta$ spanning the range from 0.5 for a hypothetical strong covalent bond character to 1.0 ionic limit. The analysis reveals that as $\beta$ increases, the energy separation between FS $J = 3$ and $J = 1$ increases, resulting in greater resolution between PE peaks A and B in the CC-RIXS map for ionic systems. In contrast, for more covalent systems ($\beta$ closer to 0.5), the energy separation between peaks A and B becomes comparable or smaller than the lifetime broadening of the $3d$ core-hole (0.8 eV[65]), the two transitions can no longer be resolved experimentally and appear as a single feature. The observed decrease in the energy separation between PE peaks A and B is therefore indicative of reduced $4f$ character in the FS, consistent with a decrease in localized $4f$ electron density in more covalent species. While Lanthanide L$_{2,3}$ edge CC-RIXS have previously been calculated[26–28,51,54,66], these efforts have largely focused on spectral assignment or electronic structure without establishing a direct connection to bonding properties. In contrast, for transition metal systems, extensive work has already demonstrated that RIXS features (transition metal K-edge or L$_{2,3}$-edge) can provide detailed insights into metal–ligand interactions, including covalency trends[67–71].

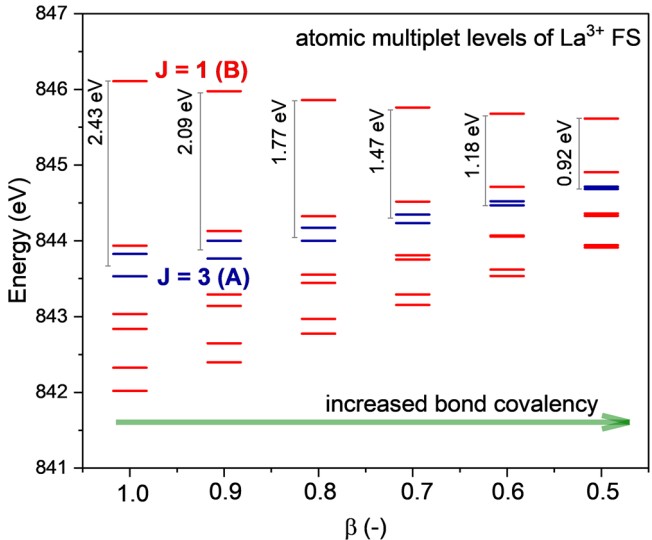

**Fig. 4 | Nephelauxetic effects on the FS electronic structure.** Changes of the atomic multiplet energy levels (red and blue bars) of $3d^9\,4f^1$ as function of the nephelauxetic ratio $\beta$ showing the energy spacing between the $J = 3$ and $J = 1$ states that are responsible for the PE peaks A and B in the CC-RIXS map, respectively. Note that only the $3d^3_{3/2}\,4f^1$ energy levels are shown for clarity.

Figure 5 shows the calculated and experimental CC-RIXS map for 9H2O, MACROPA and DOTA. The calculated maps are deconvoluted into the two main contributions ($E1_{in}$, $E1_{out}$) and ($E2_{in}$, $E1_{out}$) in the CC-RIXS process (Fig. 2). Since the complexes are non-centrosymmetric, hybridization between La $4f$ and $5d$ orbitals are possible, thereby enabling also electric-dipole ($E1_{in}$, $E1_{out}$) transitions to become active even in the PE region. In the ($E2_{in}$, $E1_{out}$) mechanism, the observed features closely resemble those of the atomic system (Fig. 3a), notably the presence of peaks A and B within the PE energy window. In the ($E1_{in}$, $E1_{out}$) mechanism, the characteristic strong WL signal dominates, but an additional weak PE feature, denoted as peak X, is also discernible. Peak A and peak X are closely overlapping in energy transfer, although peak X is slightly shifted toward higher excitation energy. The calculated branching ratio between the intensities of peaks A, B, and X is 0.47:0.17:0.36 (9H2O).

The multiplet energies of IS and FS are also shown next to the calculated maps in Fig. 5a–c. We observe only weak perturbations induced by the $4f$ ligand–field splitting on the multiplet levels, in line with the ligand–field effect of many trivalent lanthanide systems[72–76]. In fact, MACROPA has the strongest ligand–field splitting of $4f$ orbitals (0.21 eV), whereas DOTA has the strongest ligand–field splitting of $5d$ (2.50 eV) (see the Supplementary Fig. 5). For the $5d$ ligand–field splitting, we also use the second derivative of the HR-XANES spectra (*vide infra*) to deconvolute the broad WL into its constituent $2p \rightarrow 5d$ transitions, allowing the identification of inflection points associated with the $5d$ ligand–field splitting. The energy separation of these features (Supplementary Fig. 2c) shows very good agreement with the calculated $5d$ ligand–field splitting (Supplementary Fig. 5c).

Peak X originates from IS ($2p^5\,4f^1$) with higher angular momentum $J$ values (3 and 4) than those associated with peak A (see Fig. 3a), suggesting different electronic relaxation pathways. To evaluate the character of the many-electron wavefunctions, we analyze the composition of each configuration interaction eigenstate by decomposing them into their constituent Slater determinants. In particular, we focus on the degree of hybridization between La $4f$ and $5d$ orbitals in the many-electron wavefunctions. So instead of ranking the determinants by their overall weight, we separately extract and visualize those that include $4f$ and/or $5d$ spin-orbitals. This allows us to highlight the specific contributions of these orbitals to the electronic structure, and to assess their involvement in the mixing between $2p^5\,4f^1$ and $2p^5\,5d^1$ configurations in IS and between $3d^9\,4f^1$ and $3d^9\,5d^1$ configurations in FS. The resulting plots display the sum of the squared

configuration interaction coefficients ($c_i^2$) over the Slater determinants that have $4f$ (in red) and $5d$ (in black) spin-orbitals for each multiplet level $i$ (see Fig. 5a–c). This provides an orbital-specific perspective on the composition of the specific multiplet level, and reveals that hybridization between the $4f$ and $5d$ manifolds is relatively small (mixing coefficients $\leq 5\%$). This mixing is distributed across many multiplet levels but is found to be more pronounced for states associated with higher $J$ values in IS, consistent with the increased electric-dipole activity observed for peak X.

The CC-RIXS maps reveal systematic spectral shifts across the studied complexes (Fig. 5). One notable feature is the variation in energy separation between peak A and peak B in the energy transfer domain. As established in the free-ion reference study (Fig. 4), this separation reflects the strength of the interaction (in terms of nephelauxetic effect) between La $4f$ orbitals and the ligand donor orbitals. Specifically, a larger A–B separation corresponds to weaker $4f$-ligand bond covalency (i.e., more ionic character), while a smaller separation indicates increased covalent mixing. This trend is clearly observed across the series: 9H2O > DOTA > MACROPA from both theory and experiments. For the latter, Supplementary Figs. 6 and 7 show selective cross sections of the experimental CC-RIXS maps along the excitation energy and transfer energy axes, respectively. The cuts along the excitation energy are performed at three different emission energies that correspond to the maximum of the WL and pre-edge PE peaks A and B, in similar representation as in Fig. 3c for the free ion. The cuts along the transfer energy are performed at constant excitation energy for the three complexes (9H2O, MACROPA and DOTA). For 9H2O, the transfer energy cuts show a clear shoulder located 2.2 eV (PE peak B) above PE peak A (see Supplementary Fig. 7), consistent with the theoretically predicted value. For DOTA and MACROPA, theory predicts a smaller PE peaks A–B energy separation and the experimental signal-to-noise ratio is less favorable in this region; as a result, peak B cannot be identified with the same level of confidence.

An energy separation is observed in the excitation energy domain, particularly in the energy separation between WL and PE peak A, $\Delta E(5d, 4f)$. $\Delta E(5d, 4f)$ is largest in the free ion (Fig. 3), but it is progressively reduced in the complexes. This time, the trend follows 9H2O > MACROPA > DOTA. The parallel variation of these two energy separations suggests that each metric encodes complementary information about the extent of metal–ligand bonding interactions: the PE A–B peaks separation is sensitive to the $4f$ bond convalency, the PE-WL separation being sensitive to $5d$ (or both $4f$ and $5d$, vide infra). Thus, while 9H2O remains largely ionic in nature, DOTA and MACROPA display measurable covalency, albeit through different orbital channels: MACROPA promotes greater $4f$-ligand interaction, whereas DOTA facilitates enhanced mixing with $5d$ orbitals. To further examine the underlying bonding characteristics, we have collected HR-XANES spectra for all complexes.

## La L2-edge HR-XANES

Figure 6a–e shows the calculated and experimental HR-XANES signals for 9H2O, TRIS, MACROPA, DOTA, and PSMA based on the cross-sections of the CC-RIXS maps at the emission energy that corresponds to the maxima of WL. Overall, the theoretical spectra show good agreement with the experiments. Figure 6f reveals a systematic trend, where more ionic complex (example 9H2O) is characterized by large $\Delta E(5d, 4f)$ value, whereas species suspected to be more covalent (example DOTA and PSMA) exhibit smaller $\Delta E(5d, 4f)$ values. TRIS and MACROPA are found in between these two extremes. Figure 7 shows the near edge features of the HR-XANES spectra for the complexes in stacked mode. It shows that the pre-edge (PE) peaks appear at the same energy range, whereas the main absorption edges (WL) are shifted, resulting in a larger or smaller $\Delta E(5d, 4f)$. Figure 6f aslo shows that a dependence between $\Delta E(5d, 4f)$ and the electronic structure exist, which could be attributed to an underlying chemical interaction affecting both La valence $4f$ and $5d$ orbitals. The theoretical modeling also reveals that the intensity of the PE peak of the HR-XANES enclosed both ($E2_{in}$, $E1_{out}$) (peak A) and ($E1_{in}$, $E1_{out}$) (peak X) transitions with a non-trivial contribution from the ($E1_{in}$, $E1_{out}$) part (see Fig. 6a–e in the insets, the ($E1_{in}$, $E1_{out}$) and ($E2_{in}$, $E1_{out}$) contributions are depicted with the violet and orange

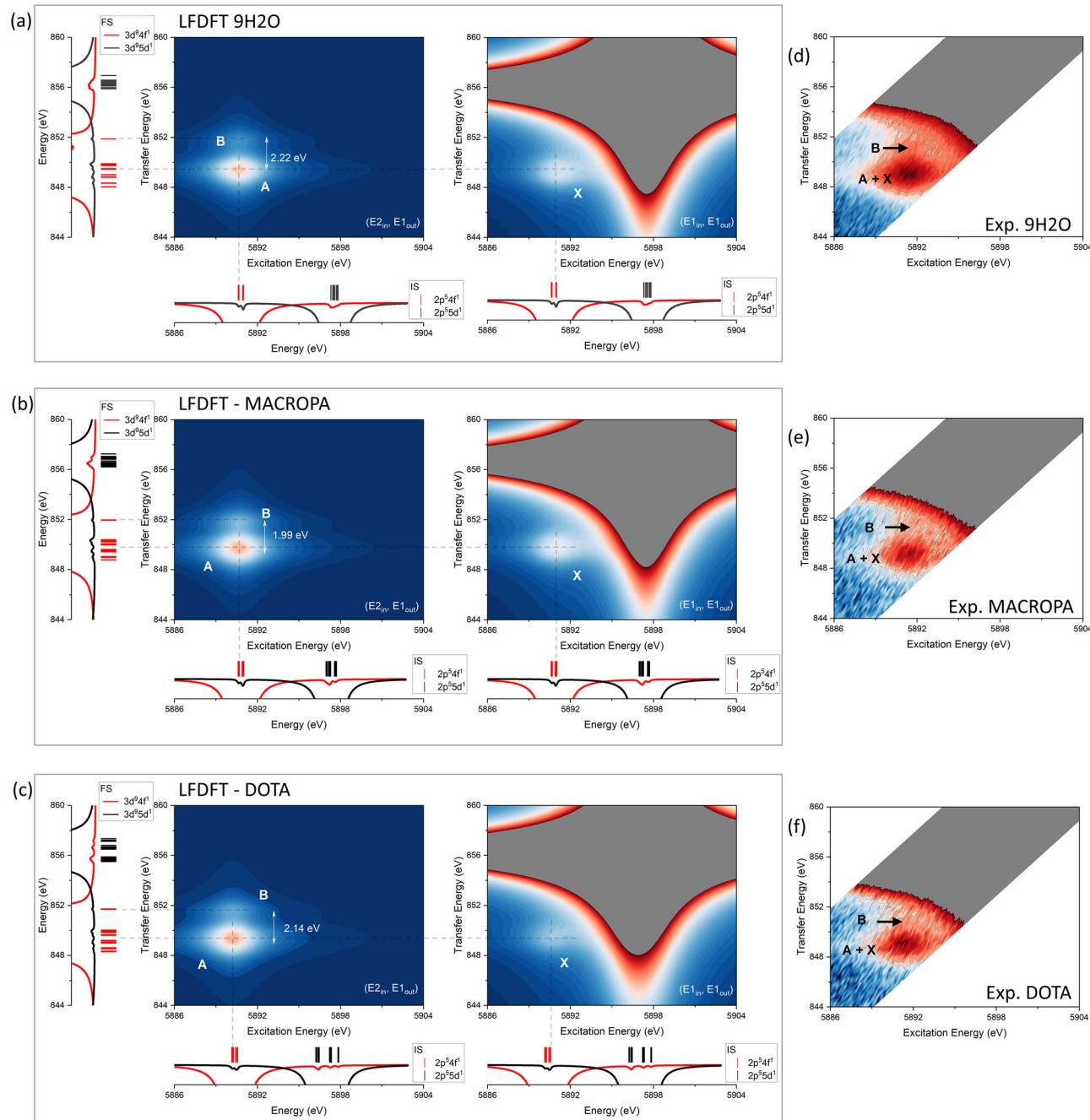

**Fig. 5 | La $L_2$-edge CC-RIXS of the complexes. a** Calculated CC-RIXS map for 9H2O. **b** MACROPA and **c** DOTA. **d** Experimental CC-RIXS map for 9H2O. **e** MACROPA and **f** DOTA. Note that the pre-edge (PE) features with multiple signal A, B and X are highlighted for both calculated (LFDFT) and experimental (exp.) data. In the calculated data, the calculated multiplet energies and ligand–field splitting of the IS and FS electronic states are also shown for discussion. These are obtained by using LFDFT with the PBE0[92] DFT functional based on the optimized structures of 9H2O, MACROPA and DOTA obtained with the same functional (Supplementary Tables 2–7). The calculated maps result from two different

mechanisms (see the text for details): $(E2_{in}, E1_{out})$ (left-side in each panel) and $(E1_{in}, E1_{out})$ (right-side). For each multiplet level ($i$), the sum of the squared of the configuration interaction coefficients ($c_i^2$) over the Slater determinants that have $4f$ (in red) and $5d$ (in black) spin-orbitals are plotted, and show a diagnostic for the degree of mixing between $2p^5 4f^1$ and $2p^5 5d^1$ configurations in IS and between $3d^9 4f^1$ and $3d^9 5d^1$ configurations in the FS. To aid visualization of the low-intensity PE features, the experimental CC-RIXS maps (**d**–**f**) are plotted on a logarithmic intensity scale. See also in the Supplementary Figs. 6 and 7, selective cross sections of the experimental CC-RIXS maps are given.

lines, respectively). This makes a direct relation between the PE intensity and $4f$ electronic structure problematic, leaving $\Delta E(5d, 4f)$ as a more reliable metric for evaluating bonding interactions.

In this context, the key question is, what quantum effects underlie $\Delta E(5d, 4f)$, making it sensitive to bond covalency? At least three primary factors contribute to $\Delta E(5d, 4f)$. First, $\Delta E(5d, 4f)$ reflects the natural energy difference between the lanthanide $4f$ and $5d$ atomic orbitals in the free atom.

For instance, in neutral La, $\Delta E(5d, 4f)$ is approximately $-2$ eV when considering it as the energy difference between the GS $5d^1 6s^2$ and the first excited-state $4f^1 6s^2$ configurations[77]. In La$^{2+}$, it is less than $-1$ eV based on the energy difference between the $5p^6 5d^1$ ground state and the $5p^6 4f^1$ excited state[78]. In La$^{3+}$, it has a small positive value[79], meaning that the unoccupied $4f$ and $5d$ are accidentally nearly degenerate, making their interplay particularly significant in the electronic structure of any La$^{3+}$ complexes.

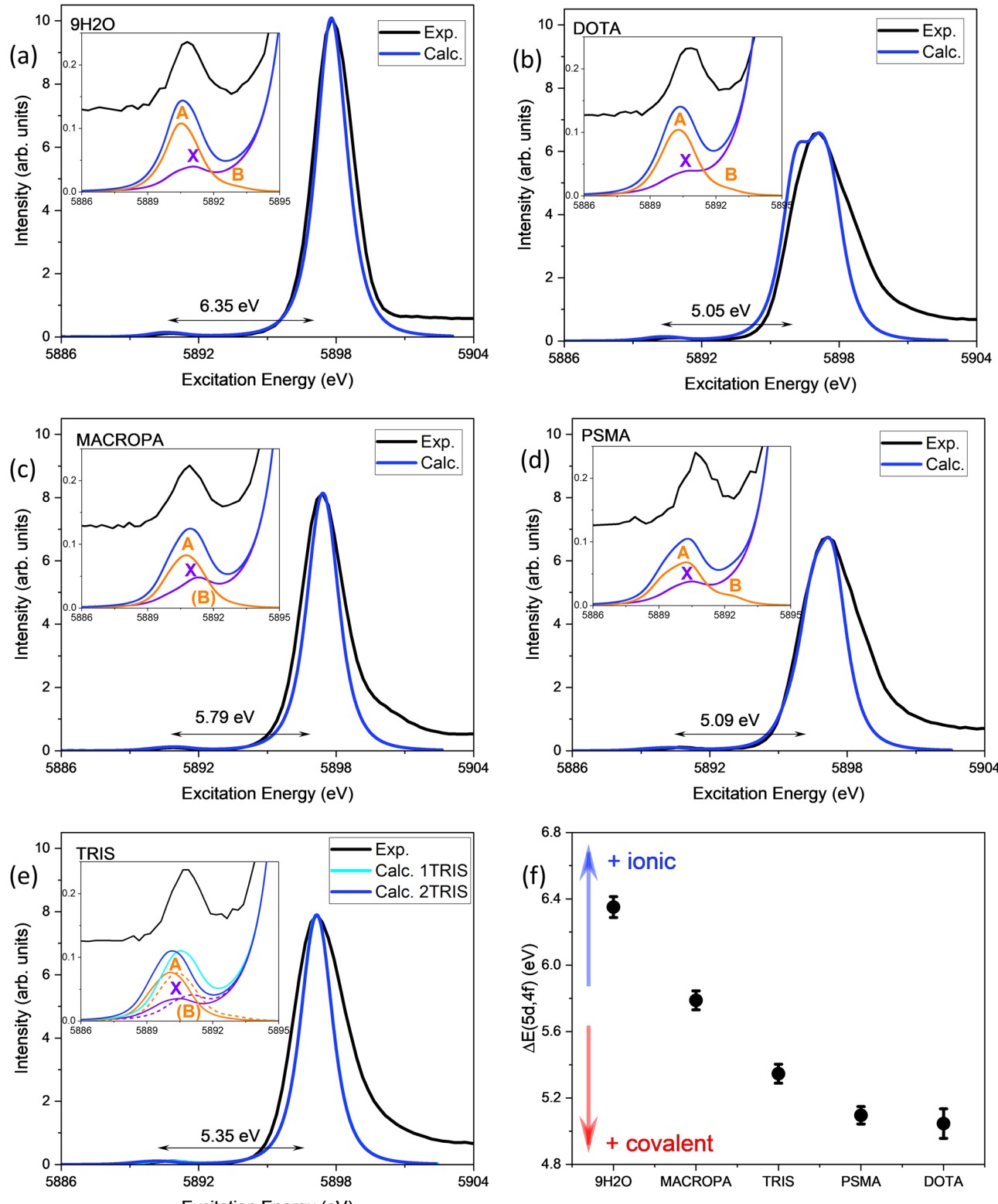

**Fig. 6 | Calculated versus experimental La L$_2$-edge HR-XANES. a** 9H2O. **b** DOTA. **c** MACROPA. **d** PSMA. **e** TRIS. The experimental spectra are depicted in black. The calculated spectra are shown in blue and cyan. **f** Correlation between the complexes and the $\Delta E(5d, 4f)$. Note that in **e** for TRIS, calculated spectra for 1TRIS (cyan) and 2TRIS (blue) are separately shown. Note also that in **a**–**e** the calculations are done using LFDFT with the PBE0[92] DFT functional based on the optimized structures obtained with the same functional (Supplementary Tables 2–7). In each panel, the

experimental derived $\Delta E(5d, 4f)$ are shown for convenience. The HR-XANES spectra are obtained from cross-sections of the calculated CC-RIXS maps at the emission energy that corresponds to the maximum of WL. The insets show highlight of the pre-edge energy region of the spectra, showing also the individual contributions of the ($E1_{in}$, $E1_{out}$) (violet) and ($E2_{in}$, $E1_{out}$) (orange) mechanisms in the CC-RIXS process. For TRIS: 1TRIS (dashed line) and 2TRIS (solid line).

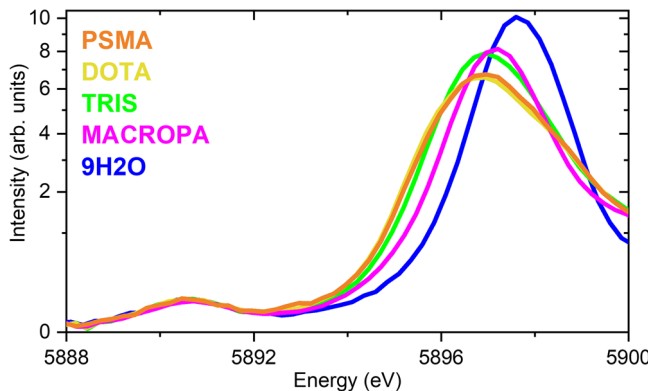

**Fig. 7 | Pre-edge La L$_2$-edge HR-XANES analysis.** The HR-XANES spectra for 9H2O, MACROPA, TRIS, DOTA, and PSMA are represented in stacked mode. Note that the intensities are represented here in logarithmic scale.

Second, $\Delta E(5d, 4f)$ incorporates the effects of ligand–field on the $4f$ and $5d$ orbitals. Namely, the spherical component of the ligand–field potential raises the metal orbital energies due to the repulsion between the $4f$/$5d$ electrons and the negatively charged ligand environment. This effect is stronger for $5d$ orbitals and weaker for $4f$ orbitals[73,76,80,81].

Finally, $\Delta E(5d, 4f)$ includes the core-hole screening effects, a key factor in X-ray spectroscopy[82]. Core-hole screening is more pronounced for $4f$ orbitals than for $5d$. Supplementary Fig. 4 shows that the energy stabilization of the overall $4f$ orbitals from GS to IS/FS ($-3.3$ eV) is twice as large as that of $5d$ ($-1.2$ eV). As a result, a larger $\Delta E(5d, 4f)$ generally corresponds to a stronger $5d$ ligand–field effect and enhanced $4f$ core-hole screening, both of which are associated with more ionic bonding interactions. Conversely, a smaller $\Delta E(5d, 4f)$ value indicates greater $5d$-ligand orbital mixing, reflecting increased covalency in the metal–ligand bond. We can therefore conclude that spectral features of the CC-RIXS and HR-XANES have highly relevant and sensitive information for bond analysis in lanthanide complexes. The present results demonstrate that these techniques can serve as complementary tools to ligand-based spectroscopies, which are often employed as the standard method for estimating bond covalency[50,83].

## Bond analysis

To further deepen our understanding of the La-ligand bonding nature, we performed theoretical bond analysis using energy decomposition analysis (EDA)[84,85] and quantum theory of atoms in molecules (QTAIM)[86]. These models give quantitative insights into electron density distribution and nature of the bonding interactions, complementing the above outlined empirical results.

Figure 8a shows EDA total interaction energy ($\Delta E_{int}$), electrostatic interaction energy ($\Delta E_{elst}$), and orbital interaction energy ($\Delta E_{oi}$) components for 9H2O, 1TRIS, 2TRIS, MACROPA, DOTA and PSMA. The $\Delta E_{int}$ data show significant differences depending on the charges of the complexes, which are determined by the charges of the ligands. Obviously, this originates from the large charge-dependence of the $\Delta E_{elst}$ component (DOTA > PSMA ≈ MACROPA > TRIS ≈ 9H2O). Variation of $\Delta E_{oi}$ covers a narrow range, where its value is somewhat larger for charged complexes. In complexes with neutral ligands (1TRIS, 2TRIS and 9H2O) the ionic and covalent contributions have similar magnitudes. These theoretical data suggest that the *TRIS* ligand is a somewhat more favorable than *H$_2$O* based primarily on its somewhat stronger covalent interactions with La, while the electrostatic interactions in the two complexes are nearly the same. This point aligns with the HR-XANES findings. As Fig. 1b reveals, the La–O distances are shorter in 1TRIS/2TRIS compared to 9H2O, suggesting stronger La-ligand interactions with the former. Due to the positive inductive effect of the alkyl moiety, the O atoms in 1TRIS are better donors (if not hindered by steric effects, like in 2TRIS) than those in 9H2O. The second *TRIS* ligand in 2TRIS increases slightly both $\Delta E_{elst}$ and $\Delta E_{oi}$

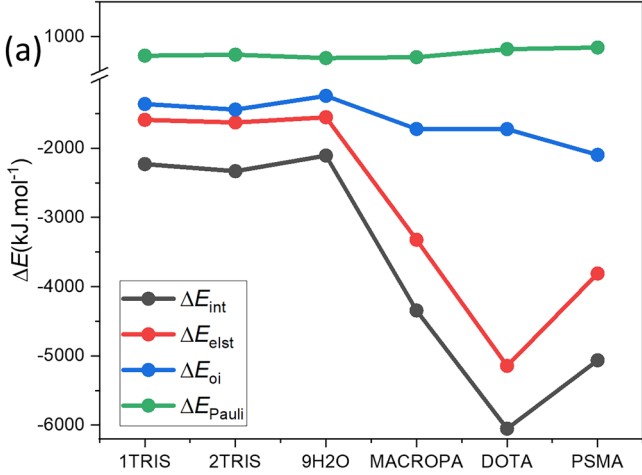

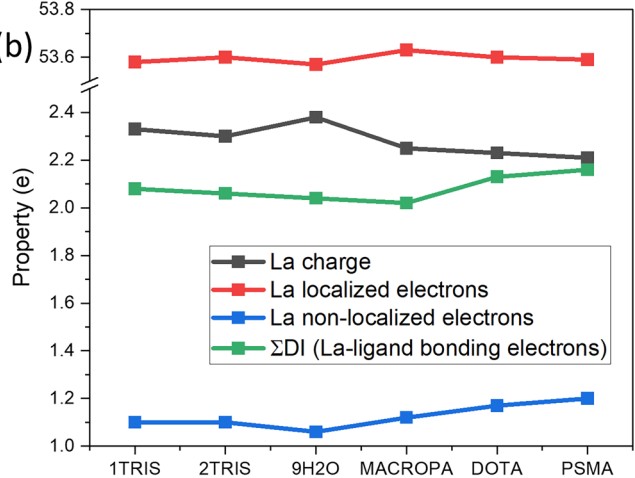

**Fig. 8 | Ab initio bond analysis. a** Energy decomposition analysis (EDA). **b** Quantum theory of atom in molecules (QTAIM) analysis. Different metrics that are used to describe chemical bonds are shown for 9H2O, 1TRIS, 2TRIS, MACROPA, DOTA, and PSMA (see the text for details). Note that the calculations are done using DFT with the TPSSh[93] DFT functional based on the optimized structures obtained with the same functional (Supplementary Tables 2–7). Description of the bond analysis metrics are available elsewhere[84,86].

contributions, thus further stabilizes the complex. The charge ($-3$ e) of the ligand in PSMA would suggests smaller $\Delta E_{int}$ and $\Delta E_{elst}$, while the integral properties from QTAIM analysis (vide infra) imply somewhat stronger covalency with respect to DOTA, also in line with the HR-XANES findings.

Figure 8b shows selected integral properties from the QTAIM analyses for 9H2O, 1TRIS, 2TRIS, MACROPA, DOTA and PSMA. The electron density assigned to La was divided to localized electrons occupying the La atomic orbitals, thus not participating in any interaction with the ligands and non-localized electrons participating in the covalent interactions with the bonding electrons from the ligands. The delocalization indices (DI) express the number of electrons forming the covalent bonds between La and the donor atoms of the ligands. As concluded from Fig. 8a, the covalent interactions and thus the above electron density properties show only small variations in the different complexes. Particularly noteworthy are the very close La charges in spite of the considerably different ligand charges. Apparently, the excess charges of the ligands in DOTA and MACROPA are well stabilized in their chemical structures, leading to only slightly larger charge transfer to La as compared with the complexes with neutral ligands. Accordingly, the localized electron density of La is nearly the same (within 0.06 e) in all complexes; the largest value is in MACROPA. The non-localized (i.e., bonding) electron density of La shows in Fig. 8b a similar trend as the localized density: larger for the negatively charged ligands,

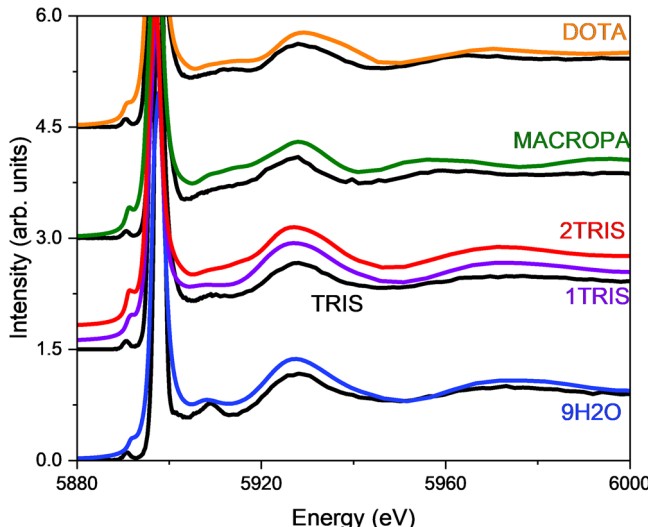

**Fig. 9 | Post-edge La $L_2$-edge HR-XANES analysis.** Comparison between the experimental (black traces) and FDMNES calculated (colored traces) HR-XANES of 9H2O, TRIS, MACROPA, and DOTA. For TRIS, the calculated spectra for 1TRIS (violet) and 2TRIS (red) are separately shown. The optimized structures obtained with PBE0[92] functional were used as input structures for the FDMNES calculation (Supplementary Tables 2–7). Note that the spectra are selectively shown in the energy region that corresponds to the HR-XANES post-edge signals.

smallest for 9H2O. In PSMA, the peptide substituent increases slightly the charge transfer to La as compared with the parent DOTA.

The covalent bonding is quantitatively described in terms of electrons by the DI values. These shared electrons show a trend of PSMA > DOTA > 1TRIS > 2TRIS > 9H2O > MACROPA. At the first sight, the small DI parameter in MACROPA seems to be in contradiction with its large $\Delta E_{oi}$ and charge transfer values. Inspecting the integral properties in more detail, we found that in MACROPA a larger part of the transferred charge is localized on La compared to the other complexes, hence less electrons from charge transfer are available for the electron sharing interactions. This peculiarity may be attributed to the mostly large La–O/N$_{ring}$ distances. However, we have to take into account that the number of shared electrons is not the only factor for the energy of the covalent interaction. This is shown for instance by the small differences in the four $\sum$DI-s, while there are significant differences in the $\Delta E_{oi}$ energies (cf. Fig. 8a). In fact, part of the transferred charge does not participate in electron sharing (reflected by DI), but is localized on La. This latter process belongs also to orbital relaxation and lowers the energy of the system, counting to $\Delta E_{oi}$.

**Coordination environment probed by HR-XANES**
Additional theoretical model is also applied to further support the analysis. Figure 9 shows calculated La $L_2$-edge HR-XANES spectra of 9H2O, 1TRIS, 2TRIS, MACROPA and DOTA obtained by means of the finite-difference method (FDMNES)[87]. FDMNES is focused on the post-edge energy region of the HR-XANES spectra, and comparison with the experiments is also provided. Since the post-edge signals reflect multiple scattering processes of the photoelectron, which are highly sensitive to the local coordination geometry, for example, bond lengths, over the overall electronic structure of the system, the model reproduces very well the experimental post-edge spectral profiles.

The FDMNES calculations were also used to clarify the stochiometry of TRIS complexes, based on a comparative theoretical analysis of the two model complexes: 1TRIS and 2TRIS, and their influence on the overall HR-XANES profiles (pre-edge plus post-edge). By comparing calculated HR-XANES spectra of the 1TRIS and 2TRIS with the experimental data, we find compelling evidence in favor of the 1TRIS form. The simulated spectrum of 1TRIS closely reproduces both the pre-edge to white-line (WL) energy separation (Fig. 5e) and the post-edge spectral profile observed

experimentally (Fig. 9). In contrast, 2TRIS yields a significantly larger pre-edge-WL energy gap and a post-edge profile that slightly diverges from the measured spectrum (additional peaks and energy shift). These differences suggest that excessive coordination by the TRIS ligand, as in 2TRIS, introduces stronger ligand-field effects that are inconsistent with the spectroscopic observations. Taken together, these findings strongly support the conclusion that only a single *TRIS* ligand is present in the coordination sphere of La$^{3+}$ under the present experimental conditions.

## Conclusions
This work demonstrates that combined experimental and theoretical approaches, spanning various level of spectroscopic methodology, provide a powerful platform to probe the chemical bonding characteristics of La$^{3+}$ complexes. Structural information was obtained via La $L_3$-edge EXAFS measurements and DFT geometry optimizations. Electronic structure analysis was performed using La $L_2$-edge CC-RIXS and HR-XANES, supported by LFDFT modeling of multiplet energy levels and X-ray spectral features. Bonding characteristics were further probed through DFT-based EDA and QTAIM analyses, which provided quantitative insights into the very nature of metal–ligand interactions. Additionally, FDMNES simulations were used to reproduce and analyze the HR-XANES post-edge energy region, offering complementary information on the structure of the complexes beyond EXAFS. We believe that the combination of these techniques has yielded a robust and coherent picture of lanthanide bonding and electronic structure in aqueous and chelated coordination environments.

Specifically, we identified two distinct features in the La $L_2$-edge CC-RIXS map and HR-XANES spectra that encode information about the extent of La $4f$ and $5d$ orbital participation in the metal–ligand bonding. Through LFDFT, we showed that both the energy separation between the two CC-RIXS PE signals correlates with the nephelauxetic effect, a central-field descriptor of orbital-specific bond covalency, here La $4f$. This analysis reveals that 9H2O exhibits ionic character, while DOTA and MACROPA show increased covalency in the La-ligand interaction. A complementary metric emerges from the HR-XANES spectra. The energy separation between WL and PE peak A ($\Delta E(5d, 4f)$) decreases along the series 9H2O > MACROPA > TRIS > DOTA $\simeq$ PSMA, further reflecting the trend toward more covalent bond character. Importantly, $\Delta E(5d, 4f)$ is sensitive to both La $4f$ and $5d$ orbitals. While HR-XANES pre-edge intensities cannot be straightforwardly interpreted due to the convolution of both the electric-dipole and electric-quadrupole transitions—particularly in non-centrosymmetric environments—our theoretical calculations indicate that the electric-quadrupole contribution is dominant. Finally, quantum chemical bonding analyses reinforce the findings, confirming the spectroscopic insights, supplementing also the proposed bond covalency trends. This underscores the utility of this combined theory-experiment framework in advancing our understanding of $f$-element bonding, but also sets the stage for future studies on actinium-based radiopharmaceuticals.

We acknowledge that direct extrapolation from La $L_2$-edge (similarly $L_3$-edge) spectra to Ac is not straightforward. Ac presents a different electronic structure, which is complicated by a different ordering and spacing between $5f$ and $6d$ orbitals and by stronger spin-orbit coupling effects and ligand–field interactions. As a consequence, the Ac $L_{2,3}$-edge spectra may exhibit weaker or altered pre-edge intensities compared with La, and the separation between features associated with $5f$ and $6d$ characters may be ambiguous. To address these differences experimentally, we propose that future Ac studies focus on the actinide $M_{4,5}$-edges (direct $3d \rightarrow 5f$ excitation) and on complementary $M_{2,3}$-edges ($3p \rightarrow 6d$ transitions), using HR-XANES and CC-RIXS techniques. Such measurements will be useful to disentangle $5f$ versus $6d$ contributions and to test whether the ligand-induced trends that we observed here for La hold for Ac.

## Methods
### Experimental La $L_3$-edge EXAFS
The La $L_3$-edge EXAFS experiments were performed at the INE-Beamline, a bending magnet beamline of the KIT Light source[88]. A Rh-coated

cylindrically bent mirror collimated the white beam, which was monochromatized using a pair of Si(111) crystals in the Lemonnier-type double crystal monochromator (DCM). The monochromatic beam was then focussed vertically and horizontally by a Rh-coated toroidal mirror. The resulting spot size was $1 \times 0.6$ mm. The samples were placed in between the first and second ionization chambers. The detector was placed at 90° angle with respect to the beam direction, at the height of the sample position. The reference material-Ti metal foil (Ti K-edge)—was placed between the second and the third ionization chambers and measured in transmission mode. The step size was determined to be 0.5 eV in the XANES region of the spectra and $k = 0.05$ Å$^{-1}$ in the post-edge and EXAFS regions.

Data analysis of the EXAFS spectra was performed by using Athena and Artemis, part of the Demeter software package (version 0.9.26)[89]. OriginLab was used for the visualization of the results. First, the spectra were calibrated by measuring the Ti K-edge on titanium metal foil as a reference. The tabulated value for the Ti K-edge (4966.4 eV[90]) was set to the first inflection point of the reference spectrum of Ti. The excitation energy of all measured spectra of the samples were corrected accordingly. Then, background corrections were preformed by fitting and subtracting a linear and quadratic functions from the pre-edge and post-edge regions of the spectra, respectively. The spectra were also normalized so that the absorption edge jump was equal to 1. From these normalized spectra, $\chi(k)$ was extracted and Fourier transformed (FT). The spectra were fitted simultaneously by weighting them, $k = 1, 2, 3$. The scattering path were calculated by using the DFT optimized structures of the complexes (Supplementary Tables 2-7). The $k$-range was set individually depending on the quality of the data. This ranged between $k = 2$ Å$^{-1}$ and $k = 8.5$ Å$^{-1}$ with a Hanning window with d$k$ of 1 (see details in Supplementary Fig. 1). The EXAFS spectra and their best fits are depicted in Supplementary Fig. 1, together with the single-scattering paths that contributed to the fit. All fit results are summarized in Supplementary Table 1 and details on the modeling strategy are given in Supplementary Section 1.3.

### Experimental La L$_2$-edge CC-RIXS and HR-XANES

The La-L$_2$ edge CC-RIXS and HR-XANES experiments were performed at the SUL-X beamline of the KIT Light Source[64]. The synchrotron radiation generated by a wiggler insertion device was passed through the Si(111) DCM and collimated using mirrors. For an energy of 5000 eV, the produced beam had a flux of about $4.10^{10}$ ph/s for 100 mA current accumulated in the storage ring. The beam was focused to about $100 \times 100$ μm$^2$ on the sample using Kirkpatrick-Baez mirrors. The SUL-X beamline operated a Johann-type Rowland-circle spectrometer in near-backscattering geometry. The sample, analyzer crystal and the detector were positioned in a Rowland circle of radius 0.5 meters. Horizontally stripped Si(331) analyzer crystal (0.5 m bending radius, ESRF) was set to 80.7° Bragg angle to monochromatise the L$\beta$1 (L$_2$-M$_4$) (5040 eV) characteristic fluorescence of La. In the near-backscattering geometry, the analyzer crystal is oriented such that the angle between the incident beam and the analyzer crystal/sample axis is 8°, corresponding to a scattering angle of about 172° $\sigma$ [29]. The scattering plane, defined by the electron bunch emitting synchrotron radiation, the sample, and the analyzer crystal, is perpendicular to the linear polarization vector of the incident beam, which lies in the plane of the synchrotron.

For the CC-RIXS experiments, the emission energy was scanned from 5034 to 5045 eV using 0.20 eV step size. The incident energy was scanned from 5885.5 to 5905.5 eV in 0.2 eV incremental steps. For the HR-XANES experiments, the emission energy was selected based on normal X-ray emission scans. Specifically, for each sample, an excitation energy of 6000 eV was used to record the normal emission spectrum over the 5032–5052 eV energy range. The maximum of the La L$_2$M$_4$ emission line (5042.4 ± 0.1 eV) was found to be identical for all the samples and was used as the fixed emission energy for the HR-XANES acquisition. For that, the excitation energy was scanned from 5733 to 6267 eV using 0.25 eV steps over the pre-edge and white line and up to 4 eV steps in the post-edge energy region. Energy calibration was performed using references Ti and Cr metal foils (at Ti and Cr K-edge), and energy stability of the spectrometer was ensured

using a La$_2$O$_3$ and a La-containing flint-stone samples that were scanned multiple times per day. Representations of the HR-XANES spectra are shown in Supplementary Fig. 2, including also the first derivative and second derivative of the signals. No radiation damage on the sample has been observed during the measurement. Supplementary Fig. 8 shows the La L$_2$-edge HR-XANES spectra for 9H2O, DOTA and MACROPA before and after the La L$_2$-edge CC-RIXS measurements.

The experimental La L$_2$-edge HR-XANES spectra were modeled using a sum of pseudo-Voigt functions. The fitting procedure was carried out using the *fminsearch* optmization routine in Matlab, which minimized the difference between the experimental and fitted spectra. The intense WL signals was reproduced using a combination of two or more functions to accurately reflect its complex shape (the energy references are taken from the second derivative signal in the Supplementary Fig. 2c). The PE energy region was sufficiently described using a single function, without optimizing the contributions from the ($E1_{in}$, $E1_{out}$) and ($E2_{in}$, $E1_{out}$) mechanisms. The fitting procedure consisted of assessing the peak locations in the data and running the program multiple times, randomly altering the fitting parameters and their ranges until a fit was found that matched the data set most accurately. The intensity and energy of the PE peaks were extracted from the fit; however, only the latter is reported in Fig. 6f. The pre-edge energies were referenced to the inflection point of the main absorption edge (maximum of the first derivative signal in the Supplementary Fig. 2b) to ensure consistency across the five different spectra.

The results of the pseudo-Voigt fitting procedure are shown in the Supplementary Fig. 3.

### Computational details

All the calculations were carried out by using DFT with the Amsterdam density functional (ADF) code in the AMS[53]. The pure DFT exchange and correlation functionals: generalized gradient approximation (GGA) via the Perdew–Burke–Erzenhof (PBE) formulation[91], as well as the hybrid PBE0[92], and the meta-hybrid Tao–Perdew–Staroverov–Scuseria (TPSSh) functionals were used[93]. Molecular orbitals were expanded in terms of all-electron Slater-type orbital (STO) basis sets of triple-zeta quality, extended with two polarization functions plus an extra $f$-STO (TZ2P+) for La, and with a single polarization function (TZP) for the light elements (C, N, O, and H)[94]. Relativistic corrections were included by using the zeroth-order regular approximation (ZORA) of the Dirac equation at both the scalar and the spin-orbit levels of theory[95,96]. Solvent effects were taken into account by using the conductor-like screening model of solvation (COSMO)[97] within the AMS program package[53]. Dispersion forces were also included by using the Grimme corrections in its D3 formulation and Becke-Johnson parametrization[98].

### Geometry optimization

The GS molecular structures of 9H2O, 1TRIS, 2TRIS, MACROPA, DOTA and PSMA (see Fig. 1a) were optimized using DFT (PBE[91], PBE0[92], and TPSSh[93] functionals). Since the complexes have closed-shell GS electronic structure, the restricted DFT formalism was always used. All structural relaxations were achieved following the standard gradient algorithm, with two times tighter convergence criteria as the default ADF values[53]. Frequency analysis was performed for the PBE geometries to confirm the minimum characters of the optimized structures.

The optimized Cartesian coordinates are listed in the Supplementary Tables 2–7 for 9H2O, 1TRIS, 2TRIS, MACROPA, DOTA, and PSMA, respectively.

### Electronic structures

The electronic structure relevant for the La L$_2$-edge CC-RIXS calculations were performed using LFDFT[52]. LFDFT has been shown to be well-suited for studying X-ray spectroscopy[82,99–102], in particular for $f$-elements coordination compounds[99,101,103–106].

In LFDFT, the average of configuration (AOC) concept were used[107]. It is based on fractional occupation numbers of selective Kohn–Sham

molecular orbitals, aiming at capturing quantum effects of systems with near degeneracy correlation and in the presence of core-hole. This allowed us to obtain state-averaged electronic densities that were isomorphic to the GS, IS, and FS electronic states relevant in the CC-RIXS process (see Fig. 2b). For GS, since there is no open-shell electrons, the reference electronic structures were obtained by using restricted SCF setup. For IS and FS, on the other hand, fractional occupations were assigned to MOs with predominant characters for La $2p$, $4f$, and $5d$ (IS) as well as $3d$, $4f$, and $5d$ (FS). These MOs were formally identified from the reference GS calculation based on their fractional parentage coefficients. These coefficients represented the portion of MOs with the most significant symmetrized fragment orbitals' (SFO) gross population for La $2p$, $3d$, $4f$, and $5d$. The coefficients were listed in the ADF[53] output file along with the molecular orbital energies.

In practice, together with the GS reference calculation, four additional single-point DFT calculations (PBE[91], and PBE0[92] functional and structures) were prepared. For IS, two calculations were required. In one input file, fractional occupation numbers were assigned such that the threefold $2p$ orbitals were populated with $5/3 = 1.6667$ electrons each and the sevenfold $4f$ orbitals with $1/7 = 0.1428$ each, reflecting the electron distribution relevant to the $2p^5 4f^1$ configuration. In another one, the threefold $2p$ orbitals were populated with $5/3 = 1.6667$ electrons each and the fivefold $5d$ orbitals with $1/5 = 0.2000$ each, representing the $2p^5 5d^1$ configuration. Similarly, for FS, two additional single-point calculations were carried out. In one calculation, the fivefold $3d$ orbitals were assigned with $9/5 = 1.8000$ electrons each, with the sevenfold $4f$ orbitals set to $1/7 = 0.1428$ each, resulting with the $3d^9 4f^1$ configuration. In the last one, the fivefold $3d$ orbitals remained occupied with $9/5 = 1.8000$ electrons each, while the fivefold $5d$ orbitals were set to $1/5 = 0.2000$ each for the $3d^9 5d^1$ configuration. These tailored occupation numbers enabled a consistent treatment of core-valence interactions important in the CC-RIXS process. Examples of electronic structures (density of states (DOS) and angular momentum projected DOS as function of GS, IS, and FS) are shown in the Supplementary Fig. 4.

### LFDFT calculation of RIXS signals

Ligand–field parameters were derived from the electronic structures (see above). These parameters included the state-averaged configuration energy parameters, the Slater–Condon integrals, the spin-orbit coupling constants, and the matrix elements of the ligand–field potential[108].

In total, five state-averaged configuration energy parameters were obtained: $\Delta E(4f^0)$, $\Delta E(2p^5 4f^1)$, $\Delta E(2p^5 5d^1)$, $\Delta E(3d^9 4f^1)$, and $\Delta E(3d^9 5d^1)$. They corresponded to the computed DFT total electronic energies of the reference GS calculation, as well as of the four other excited-states calculations (two IS and two FS). Sixteen Slater–Condon integrals were also obtained: $F^2(2p,4f)$; $G^k(2p,4f)$ $k = 2, 4$; $F^2(2p,5d)$; $G^k(2p,5d)$ $k = 1, 3$; $F^k(3d,4f)$ $k = 2, 4$; $G^k(3d,4f)$ $k = 1, 3, 5$; $F^k(3d,5d)$ $k = 2, 4$; and $G^k(3d,5d)$ $k = 0, 2, 4$. They were calculated based on the radial functions of the La $2p$, $3d$, $4f$, $5d$ orbitals, which were extracted from the IS and FS DFT calculations[108]. Four spin-orbit coupling constants, $\zeta_{2p}$, $\zeta_{3d}$, $\zeta_{4f}$ and $\zeta_{5d}$, were also calculated based on the relativistic ZORA spin-orbit method. A matrix with twenty times twenty elements were also obtained, which represented the ligand–field potential for the open-shell $2p$, $3d$, $4f$, and $5d$ electrons.

The calculated ligand–field parameters are listed in the Supplementary Table 8 and compared with some reference values for La$^{3+}$ [28]. The ligand–field potential for the $4f$ and $5d$ electrons are represented in Supplementary Fig. 5. These parameters were used to set up the effective Hamiltonian[82,108] to calculate the multiplet energies and ligand–field wavefunctions for GS, IS and FS, and to model the CC-RIXS spectra.

The CC-RIXS processes were simulated using the Kramers-Heisenberg formula, and explicitly accounting for the SUL-X beamline scattering geometry (for details, see also in the Supplementary Theoretical Information).

### Bond analysis
**EDA.** The energetic characteristics of La-ligand interaction were assessed by EDA[84,85] at the DFT level of theory (PBE[91], PBE0[92], and TPSSh[93] functionals and structures). The different DFT exchange correlation

functionals were probed to verify the reliability of the metrics for the trends in the bonding properties. There were small systematic differences between the results obtained by the three functionals, but the trends were consistent. We selected the TPSSh[93] functional for a detailed analysis of bonding in the complexes (see Fig. 8a).

In the analyses, two fragments were considered: the La$^{3+}$ ion as one fragment and all the ligands together as the second fragment. The interaction energy between the two fragments, $\Delta E_{int}$, is defined as $\Delta E_{int} = \Delta E_{elst} + \Delta E_{Pauli} + \Delta E_{oi}$, where $\Delta E_{elst}$ corresponds to the classical electrostatic interaction between the charge distributions of the isolated fragments after brought together in the complexes, $\Delta E_{Pauli}$ is the repulsion between occupied orbitals (corresponding to the steric repulsion) and $\Delta E_{oi}$ is the orbital interaction energy between the fragments in the complexes, accounting for electron pair bonding, charge transfer and polarization[109].

**QTAIM.** La-ligand bonding properties were also analysed by quantum theory of atoms in molecules (QTAIM)[86] at the DFT level of theory. Similar to EDA, the PBE[91], PBE0[92], and TPSSh[93] exchange correlation functionals were used giving only rise to small systematic differences between the three functionals. We selected the TPSSh[93] functional for a detailed analysis of bonding in the complexes (see Fig. 8b). The electron density $\rho(r)$, the localization $(\lambda(r))$ and the delocalization $(\delta(r))$ indexes were reported.

### FDMNES calculations
The calculation of the La L$_2$-edge XANES spectra of 9H2O, 1TRIS, 2TRIS, MACROPA, and DOTA were obtained with the FDMNES code[87] using the finite difference method. The optimized DFT structures were used as input for the calculation. The electronic structure was calculated using DFT with a local exchange-correlation potential. Relativistic effects were included in the calculation using the Dirac-Slater approach, and spin-orbit coupling was taken into consideration for the core-electrons. Electric-quadrupole transitions were also included in the simulation. The calculated spectra were broadened with the Lorentzian function (half-width at half maximum of 1.50 eV) to simulate the $2p$ core-hole lifetime line broadening, as well as Gaussian function (half-width at half maximum of 0.25 eV) to simulate the experimental broadening. We used the default FDMNES parameter values for the step functions that mimic the absorption jump of the La L$_2$-edge. Structure of the FDMNES input file is available in the Supplementary Table 9.

### Data availability
The datasets generated during and/or analysed during the current study are available in the KIT-open repository (https://www.bibliothek.kit.edu/english/kitopen.php) with the identifier(s) [https://radar.kit.edu/radar/en/dataset/us2796rbe0wzwv9u] .

### Code availability
LFDFT is available in the Amsterdam modeling suite codes and can be obtained from https://www.scm.com.

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

## Acknowledgements

The authors gratefully acknowledge the funding from the Deutsche Forschungsgemeinschaft (DFG) through the Collaborative Research Centre 4f for Future project A1 and A3 (CRC 1573, project number 471424360); from the Federal Ministry of Education and Research (BMBF) and the Baden-Württemberg Ministry of Science as part of the Excellence Strategy of the German Federal and State Governments; and the European Research Council under the European Union's Horizon 2020 research and innovation program (grant agreement number 101003292). The authors acknowledge support by the state of Baden-Württemberg through bwHPC and the DFG through grant number INST 40/575-1 FUGG (JUSTUS 2 cluster). We also thank the Institute for Beam Physics and Technology (IBPT) for the operation of the storage ring, the Karlsruhe Research Accelerator (KARA).

## Author contributions

T.V., B.S., M.B.S., M.W.H., and H.R. conceptualized the project and designed the overall experimental and theoretical strategies; T.V. supervised data analyses and interpretation; M.S. and M.B.S. synthesized the ligands and prepared the complexes; P.V.M., B.S., and K.D. performed EXAFS experiments and analyzed EXAFS data; E.M.R., C.R., R.S.K.E., J.G., and R.S. carried out and analyzed CC-RIXS and HR-XANES measurements (9H2O, DOTA and MACROPA); P.V.M., B.S., J.G., and R.S. performed HR-XANES measurements (TRIS and PSMA); H.R., M.T., and M.W.H. designed the theoretical spectroscopic model; H.R. designed the LFDFT model and carried out LFDFT calculations; H.R. and A.K. performed the DFT calculations and bond analysis; H.R. drafted the manuscript with input from all authors. All authors discussed the results and approved the final version of the manuscript.

## Funding

## Competing interests

The authors declare no competing interests.
