## [Transparent Peer Review File · Communications Chemistry]

High-energy resolution X-ray spectroscopy reveals bonding characteristics of La³⁺ homologues of actinium radiopharmaceuticals

Corresponding Author: Professor Tonya Vitova

Version 0:

Reviewer comments:

Reviewer #1

(Remarks to the Author)
attached

Reviewer #2

(Remarks to the Author)

Core-to-core resonant inelastic x-ray scattering and high-energy resolution x-ray absorption near-edge structure techniques were used to probe metal-ligand interactions between La and various ligands. Key information regarding ligand-field effects and bond covalency was discussed. Quantum chemical modeling is included to provide a deeper analysis of the collected data.

An additional proofreading is required to catch typographical errors.

Overall, this was a nicely written body of work with a sufficient level of explanation. The only improvement suggested is the extension of this work to Ac more clearly described. It is not immediately apparent how this will translate to Ac work. The 5f elements do not mirror the 4f elements regarding energy levels.

This manuscript should be published with minor revisions.

Version 1:

Reviewer comments:

Reviewer #1

(Remarks to the Author)

I greatly appreciate the authors' efforts in addressing all of my comments. The manuscript has been significantly improved and is now ready for publication. I would like to congratulate the authors once again on this excellent work.

Response to the reviewer's comments

We are thankful to the Reviewers for providing comments and suggestions to improve the quality of the manuscript. Based on all the comments and suggestions given, the manuscript has been revised. All modifications introduced in the revised manuscript are marked in blue for convenience.

In the following we address the Reviewer's concerns point by point:

Reviewer 1

Recommendation — The Harry Ramanantoanina and co-authors report an interesting study on La-based compounds relevant to pharmaceutical applications. The authors employ advanced CC-RIXS methods and state-of-the-art electronic structure calculations to reveal the nature of chemical bonds between various compounds. The manuscript is well written and will clearly attract significant attention from the community working in the fields of actinides and lanthanides.

Response: We are grateful to the Reviewer for their suggestions and comments. The authors' responses to these comments are reported below:

Comment 1 — However, I have several questions, and some clarifications and verifications should be added to the article before it can be accepted for publication:

Why did the authors use the La L₂ edge instead of the La L₃ edge? Does the choice depend on the crystal analyzers available?

Response: The two intense emission lines L_{β1} and L_{α1} could be measured with analyser crystals providing large Bragg angle, which is needed for high experimental energy resolution. However, at the time of the experiments, we had only Si(331) analyzer crystal available and therefore we chose to study LaL₂-edge instead of L₃-edge experiments. It is interesting to compare to the L₃-edge edge and we will do this in future studies (see below Table RL1).

Table RL1: Characteristics of the spectrometer for the L_{2,3}-edges HR-XANES and CC-RIXS experiments: excitation and emission lines, energy (in eV), probability of the emission (in %) are taken from [<https://doi.org/10.1107/S0909049505012719>], together with the analyzer crystal and calculated Bragg angle (in °).

Excitation		Emission				
Edge	Energy	Edge	Probability	Energy	Analyzer Crystal	Bragg Angle
L ₂	5891	L _{β1}	85.01	5038	Si(331)	80.97
L ₃	5483	L _{α1}	75.20	4647	Si(400)	79.28
		L _{β2}	15.54	5378	Ge(422)	86.54

Comment 2 — The analysis of bond lengths and the comparison between EXAFS and DFT-based calculations are excellent. Bravo for authors! I wish more studies reported similar results, since this forms the foundation of all such analyses.

Response: Thanks, for this positive feedback. We are pleased that our combined EXAFS and DFT analysis was appreciated, and agree that such comparative approaches are essential for establishing reliable structural interpretations.

Comment 3 — Why is 2p labeled as $2p^1$ and 3d as $3d^1$? This notation suggests that there is one electron in the 2p shell in the intermediate state, which I believe is incorrect (also in the text). The transition should be $2p^64f^0 \rightarrow 2p^54f^1$. Similarly, for 3d, the correct description is $3d^{10} \rightarrow 3d^95d^1$. Unless the authors mean something different by $2p^1$ and $3d^1$, this point should be clarified to avoid confusion.

Response: We thank the Reviewer for pointing this out and appreciate the opportunity to clarify our notation. The described transitions correspond to the following: $2p^64f^0 \rightarrow 2p^5(4f,5d)^1$ and $3d^{10}4f^0 \rightarrow 3d^9(4f,5d)^1$. In the manuscript, we added the spin-orbit index specifying from which core subshell the excitation originates, i.e. $2p_{1/2}$ and $3d_{3/2}$. We will revise the text to make this notation clearer and avoid confusion.

In the revised manuscript,

- On Page 3, we have changed the notations in the following paragraph: "Figure 2b shows the electronic structures that are involved in CC-RIXS. The ground-state (GS) of the La^{3+} complexes is characterized by a formal $4f^0$ configuration resulting with a singlet electronic state derived from the $J = 0$ (1S) atomic multiplet term. The intermediate state (IS) involves electronic configurations $2p^54f^1$ and $2p^55d^1$, namely the spin-orbit components $2p_{1/2}^14f^1$ and $2p_{1/2}^15d^1$. The final state (FS) corresponds to $3d^94f^1$ and $3d^95d^1$ configurations. The CC-RIXS process is governed by two main contributions. First, the GS ($4f^0$) \rightarrow IS ($2p^55d^1$) \rightarrow FS ($3d^95d^1$) mechanism is a process with two photons described by two consecutive electric-dipole transitions ($E1_{in}$, $E1_{out}$). Second, the GS ($4f^0$) \rightarrow IS ($2p^55d^1$) \rightarrow FS ($3d^95d^1$) mechanism is a process described by electric-quadrupole and electric-dipole transitions ($E2_{in}$, $E1_{out}$). The LFDFT method enables the calculation of the multiplet energy levels and ligand-field splitting associated with IS and FS from first principles, and the simulation of the corresponding CC-RIXS maps based on the Kramers-Heisenberg formula."
- On Page 4, we have changed the notations in the following paragraph: "Additionally, Figure 3a displays the calculated atomic multiplet energy levels associated with ($2p^54f^1$, and $2p^55d^1$) as well as FS ($3d^94f^1$, and $3d^95d^1$) electronic structures. These multiplet energy levels are aligned with the CC-RIXS map, enabling a direct correlation between individual CC-RIXS features and the underlying electronic structure."
- On Page 4, we have changed the notations in the following paragraph: "The WL region exhibits a dominant signal centered at an excitation energy of approximately 5915.8 eV, corresponding to a principal peak at a transfer energy of 854.3 eV, accompanied by two satellite features at lower transfer energies. These features are attributed to IS with $J = 1$ ($2p^55d^1$), decaying into final states (FS) with $J = 0$ and $J = 2$, associated with $3d^95d^1$ multiplet structures. In the PE region, three distinct signals are identified at an excitation energy of around 5905.5 eV. The most intense peak (denoted as peak A in Figure 3a) appears at a transfer energy of 843.3 eV, while two weaker features (peaks B and C) are observed at 846.1 eV and 829.3 eV, respectively. Analysis of the multiplet energy levels reveals that these PE features originate from IS with $J = 2$ ($2p^54f^1$),

decaying into FS with $J = 3$ and $J = 1$ ($3d^9 4f^1$). Peaks A and B arise from $3d_{3/2}^3 4f^1$, peak C with higher energy transfer arises from $3d_{5/2}^5 4f^1$, which is reached with weak intensity because of the spin-orbit coupling of the La 3d electrons. Thus the CC-RIXS analysis shows that the three PE signals (A, B and C) originate from intra-atomic electron-electron interaction in FS and the spin-orbit coupling of the 3d electrons. The calculated branching ratio A:B:C is found to be 0.57:0.41:0.02 for the atomic system.”

- On page 6, we have changed the notations in the following paragraph: “Peak X originates from IS ($2p^5 4f^1$) with higher angular momentum J values (3 and 4) than those associated with peak A (see Figure 3a), suggesting different electronic relaxation pathways. To evaluate the character of the many-electron wavefunctions, we analyze the composition of each configuration interaction eigenstate by decomposing them into their constituent Slater determinants. In particular, we focus on the degree of hybridization between La 4f and 5d orbitals in the many-electron wavefunctions. So instead of ranking the determinants by their overall weight, we separately extract and visualize those that include 4f and/or 5d spin-orbitals. This allows us to highlight the specific contributions of these orbitals to the electronic structure, and to assess their involvement in the mixing between $2p^5 4f^1$ and $2p^5 5d^1$ configurations in IS and between $3d^9 4f^1$ and $3d^9 5d^1$ configurations in FS. The resulting plots display the sum of the squared configuration interaction coefficients (c_i^2) over the Slater determinants that have 4f (in red) and 5d (in black) spin-orbitals for each multiplet level i (see Figure 4a-c). This provides an orbital-specific perspective on the composition of the specific multiplet level, and reveals that hybridization between the 4f and 5d manifolds is relatively small (mixing coefficients $\leq 5\%$). This mixing is distributed across many multiplet levels but is found to be more pronounced for states associated with higher J values in IS, consistent with the increased electric-dipole activity observed for peak X.”

Comment 4 — The same issue appears in Fig.3: the figure shows $2p^1$, while the annotation refers to $2p^5$.

Response: We will also revise the figure to make the notation clearer and avoid confusion. See also comment 3.

Figure RL1: Changes in Figure 3

Comment 5 — The discussion on how reduction of Slater integrals influences the spectral shape is very valuable and very nice. However, the following statement is misleading: “In contrast, for more covalent systems (β closer to 0.5), the two peaks converge and may appear as a single unresolved feature due to the 3d core-hole lifetime broadening.” The convergence does not occur because of the 3d core-hole lifetime broadening. Please reformulate this sentence.

Response: We agree that the convergence of the two emission lines arises from the reduced multiplet splitting as the covalency increases (i.e., smaller Slater integrals with higher β), not from the 3d core-hole lifetime broadening. However, when the resulting energy separation becomes comparable to or smaller than the lifetime broadening, the two transitions can no longer be resolved experimentally and appear as a single feature. We have revised the text accordingly.

In the revised manuscript, we have rephrased the sentence as follows (Page 6): **In contrast, for more covalent systems (β closer to 0.5), the energy separation between peaks A and B becomes comparable or smaller than the lifetime broadening of the 3d core-hole, the two transitions can no longer be resolved experimentally and appear as a single feature.**

Comment 6 — Figure 4: I am surprised that RIXS experimental data do not show any splitting in the pre-edge structure (in the energy transfer direction), whereas the theoretical analysis predicts a splitting of ≈ 2 eV. The experimental resolution should be sufficient to resolve this feature. Why do the authors not observe it? (CeO_2 has also $4f^0$ ground state and pre-edge splits in a similar way and well seen in a data). Perhaps using a logarithmic scale in the 2D RIXS plots or cuts through the RIXS map in the pre-edge region would help.

Response: Following the Reviewer recommendations, we have redesign the experimental data plots. In the revised manuscript, the data are now represented on a logarithmic intensity scale in Figure 4 d-f

(see also below Figure RL2). In the present case, the weak higher-energy component becomes more apparent. Plus, in the Supplementary Information, we have prepared Figure S7 and S8 as cross sections of the experimental CC-RIXS maps along the excitation and transfer energies (see also below Figures RL3 and RL4). These selective cuts also help defining the two PE peaks (A+X and B) in the 9H2O, MACROPA and DOTA complexes.

Figure RL2: Updated figure 4 d-f (from left to right)

Figure RL3: New figure in the Supplementary Information Figure S7

In the revised manuscript, we have added the following text in the Figure 4 caption: "To aid visualisation of the low-intensity PE features, the experimental CC-RIXS maps (d-f) are plotted on a logarithmic intensity scale. See also in the Supplementary Information Figure S7 and Figure S8, selective cross sections of the experimental CC-RIXS maps are given."

Figure RL4: New figure in the Supplementary Information Figure S8

In the revised manuscript, we have added the following text on page 6: For the latter, Supplementary Information Figure S7 and Figure S8 show selective cross sections of the experimental CC-RIXS maps along the excitation energy and transfer energy axes, respectively. The cuts along the excitation energy are performed at three different emission energies that correspond to the maximum of the WL and pre-edge PE peaks A and B, in similar representation as in Figure 3c for the free ion. The cuts along the transfer energy are performed at constant excitation energy for the three complexes (9H₂O, MACROPA and DOTA). For 9H₂O, the transfer energy cuts show a clear shoulder located 2.2 eV (PE peak B) above PE peak A (see Supplementary Information Figure S8), consistent with the theoretically predicted value. For DOTA and MACROPA, theory predicts a smaller PE peaks A–B energy separation and the experimental signal-to-noise ratio is less favorable in this region; as a result, peak B cannot be identified with the same level of confidence.

Comment 7 — In fact, the experimental data shown in Fig. 4d–f do not display any differences in the pre-edge region between DOTA, MACROPA, and 9H₂O compounds. Is this related to the use of a 0.5 m spectrometer? (Although I doubt this, since the Bragg angle of the analyzer is close to 80°, and the resolution should be quite good.) I would like to hear the authors' thoughts on this point.

Response: See also Response-to-Comment 6.

Although the overall pre-edge spectral shapes of the three La³⁺ complexes appear broadly similar (see Figure 4d–f), the newly added cuts through the CC-RIXS maps (Supplementary Information Figures S7 and S8) reveal subtle but measurable differences. For 9H₂O, the transfer energy cuts show a clear shoulder located 2.2 eV (PE peak B) above PE peak A (Figure S8), consistent with the theoretically

predicted value. For DOTA and MACROPA, theory predicts a smaller PE peaks A–B energy separation and the experimental signal-to-noise ratio is less favorable in this region; as a result, peak B cannot be identified with the same level of confidence.

As requested by the Reviewer, we provide here our thoughts for using the 0.5 m spectrometer. The measured elastic peak on Teflon yields an experimental energy resolution of 1.26 eV, which is effectively comparable to what is typically achieved with a 1.0 m spectrometer under similar conditions. As the Reviewer pointed out, at high Bragg angles (close to 80°) the change in energy resolution (0.5 m versus 1.0 m) is expected to be minimal. For reference, our estimate of the expected energy resolution for a typical 1.0 m spectrometer yields a value of 1.16 eV (this is obtained by combining an analyzer crystal contribution of 0.38 eV with a monochromator contribution of 1.1 eV). This estimate is based on a 1.0 m spectrometer using a Si 111 DCM and a Si 400 analyzer crystal, operating at a high Bragg angle (79.28°) and an energy of 4839.2 eV. This shows that the performance of our 0.5 m setup is well within the operating range and we do not expect significant improvements by changing to 1.0 m geometry.

Comment 8 — The authors noticed in their theoretical simulations that the shape of the pre-edge depends on the choice of emission energy. If so, why were HR-XANES experiments not shown at least at two different emission energies? (Fig. 5 shows an experimental pre-edge structure that appears as a single peak, whereas Fig. 3 reports theoretical results showing a triple pre-edge structure at certain emission energies.) Can the authors extract such features from the RIXS maps by making cuts through the recorded data (or by analyzing single HR-XANES spectra at different emission energies, if recorded with step size)?

Response: See also Response-to-Comments 6 and 7.

Following the suggestion of the reviewer, we prepared a new figure in the Supplementary Information, Figure S7 (see also Figure RL3 above), which shows selected cross-sections of the CC-RIXS maps of 9H₂O, MACROPA, and DOTA in the same representation as Figure 1c for the free ion La³⁺ system. These cross-sections are taken at emission energies corresponding to the maximum of the WL, as well as to the maxima of the pre-edge peaks A and B. As discussed above, the pre-edge features A and B are not that well resolved as in the theoretical spectrum; therefore, clearly separated multiple peaks are not visible like in the theoretical spectra. In addition, the signal-to-noise ratio is low in the pre-edge region of the DOTA and MACROPA spectra, making quantitative analysis difficult. Even though the merging of the A and B pre-edge peaks is only barely visible for the CC-RIXS spectra of DOTA and MACROPA, we believe that the effect is present in the experimental data (see the CC-RIXS maps in log scale in the main text Figure 4d-f). Future studies of additional systems will verify this findings.

Comment 9 — Fig. 5 shows the separation between the pre-edge and main edge peak is ≈ 5 –6 eV, depending on the compound. What is not clear to me is whether the maximum of the white line is shifted, or whether the pre-edge is shifted. Can the authors show experimental HR-XANES data for all compounds plotted together (e.g., in stacked mode)? Although Fig. 7 presents experimental data, it focuses on post-edge features, making it hard to assess shifts. Similarly, Fig. S2 shows first derivatives and inflection points of the white line, indicating a shift of the white line between compounds. But what about the pre-edge — does it shift or remain constant? Clarification would greatly help readers and future researchers if the same methodology will be applied.

Response: Following the comment, we included in the revised manuscript a new Figure: Figure 6 showing the experimental HR-XANES spectra of all the complexes plotted together in a stacked mode

(see also below Figure RL6). This comparison clearly shows that the pre-edge feature appears at the same energy for all compounds, while the observed variation in the pre-edge to white-line separation arises from a systematic shift of the white-line maximum.

Figure RL5: New Figure 6 is introduced in the main text

On page 8, the following text is added in the revised manuscript: Figure 6 shows the near edge features of the HR-XANES spectra for the complexes in stacked mode. It shows that the pre-edge (PE) peaks appear at the same energy range, whereas the main absorption edges (WL) are shifted resulting in a larger or smaller $\Delta E(5d,4f)$.

Comment 10 — Finally, I did not find in SI on how was the emission energy selected for the HR-XANES measurements shown in Fig. 5? Was it fixed for all compounds, or varied? (I understood that maximum of Exes has been chosen, but this maximum remained constant or not for different La-compounds)

Response: For each sample, the excitation energy was first set at 6000 eV, and an emission energy scan was performed in the energy range 5032–5052 eV to record the non-resonant x-ray emission spectrum (see for instance in Figure RL4 the emission spectra of DOTA and MACROPA). The maximum of the emission L_2M_4 line was then determined from this spectrum and found to be identical within error for all investigated compounds, at 5042.4 ± 0.1 eV. This energy value was then chosen as the emission energy for all subsequent HR-XANES measurements.

Description of this procedure have been added to the Methods section (Supplementary Information section "Experimental La L_2 -Edge CC-RIXS and HR-XANES"). Namely, we rephrased the paragraph and included details about the emission energy: "For the HR-XANES experiments, the emission energy was selected based on non-resonant x-ray emission scans. Specifically, for each sample, an excitation

energy of 6000 eV was used to record the non-resonant emission spectrum over the 5032–5052 eV energy range. The maximum of the La L_2M_4 emission line (5042.4 ± 0.1 eV) was found to be identical for all the samples and was used as the fixed emission energy for the HR-XANES acquisition. For that, the excitation energy was scanned from 5733 to 6267 eV using 0.25 eV steps over the white line and up to 4 eV steps in the pre-edge and post-edge energy region.”

Figure RL6: Normal emission spectra of DOTA and MACROPA obtained at excitation energy of 6000 eV.

Comment 11 — I particularly enjoyed the section on “what quantum effects underlie $\Delta E(5d,4f)$, making it sensitive to bond covalency?” and thank the authors for this insightful analysis.

Response: Thanks!

Comment 12 — It is very encouraging to see that QTAIM and bond analysis align well with the trends observed in CC-RIXS. The explanations are detailed and clear, and I appreciate the effort the authors put into this section.

Response: Thanks!

Comment 13 — Last but not least, while the study presents a great analysis of La³⁺-based compounds, I would advise caution with the statement: “This work also lays the foundation for future studies on actinium complexes.” Actinium has the ground state $6d^1 7s^2$, so Ac³⁺ has neither 5f nor 6d states (similar to Th). Consequently, the Ac L₃ edge will exhibit a very different spectral profile compared to La L₃. The pre-edge might be almost negligible, and the separation between 5f and 6d states will not be noticeable. Moreover, the nature of the pre-edge at the An L₃ edge is still debated (whether it is purely 2p–5f transitions or not). I expect that L₃ CC-RIXS will be very different between Ac and La, and perhaps authors can mention it in their manuscript.

Response: We thank the reviewer for this important clarification. We fully agree that Ac will show different L-edge spectral fingerprints than lanthanum because of the different ordering and occupancy of the valence shells, and that the detailed nature and intensity of any pre-edge at an actinide L_{2,3}-edges remain debated. We have therefore softened the original statement and added a paragraph in the Discussion (see below) explaining these differences and proposing experimentally accessible actinium probes (M-edge spectroscopy and high-resolution methods) together with the required theoretical treatment.

The following paragraph has been added in the revised manuscript in the conclusion section: We acknowledge that direct extrapolation from La L₂-edge (similarly L₃-edge) spectra to Ac is not straightforward. Ac presents a different electronic structure, which is complicated by a different ordering and spacing between 5f and 6d orbitals and by stronger spin–orbit coupling effects and ligand-field interactions. As a consequence, the Ac L_{2,3}-edge spectra may exhibit weaker or altered pre-edge intensities compared with La, and the separation between features associated with 5f and 6d characters may be ambiguous. To address these differences experimentally, we propose that future Ac studies focus on the actinide M_{4,5}-edges (direct 3d → 5f excitation) and on complementary M_{2,3}-edges (3p → 6d transitions), using HR-XANES and CC-RIXS techniques. Such measurements will be useful to disentangle 5f versus 6d contributions and to test whether the ligand-induced trends that we observed here for La hold for Ac.

Reviewer 2

Recommendation — Core-to-core resonant inelastic x-ray scattering and high-energy resolution x-ray absorption near-edge structure techniques were used to probe metal-ligand interactions between La and various ligands. Key information regarding ligand-field effects and bond covalency was discussed. Quantum chemical modeling is included to provide a deeper analysis of the collected data.

An additional proofreading is required to catch typographical errors.

Overall, this was a nicely written body of work with a sufficient level of explanation. The only improvement suggested is the extension of this work to Ac more clearly described. It is not immediately apparent how this will translate to Ac work. The 5f elements do not mirror the 4f elements regarding energy levels.

This manuscript should be published with minor revisions.

Response: We are grateful to the Reviewer for their suggestions and comments.

Similar to the last comment of Reviewer 1, we added in the revised manuscript the following statement to describe the compatibility of our method to Ac studies: in the conclusion section: **We acknowledge that direct extrapolation from La $L_{2,3}$ -edge (similarly $L_{2,3}$ -edge) spectra to Ac is not straightforward. Ac presents a different electronic structure, which is complicated by a different ordering and spacing between 5f and 6d orbitals and by stronger spin-orbit coupling effects and ligand-field interactions. As a consequence, the Ac $L_{2,3}$ -edge spectra may exhibit weaker or altered pre-edge intensities compared with La, and the separation between features associated with 5f and 6d characters may be ambiguous. To address these differences experimentally, we propose that future Ac studies focus on the actinide $M_{4,5}$ -edges (direct $3d \rightarrow 5f$ excitation) and on complementary $M_{2,3}$ -edges ($3p \rightarrow 6d$ transitions), using HR-XANES and CC-RIXS techniques. Such measurements will be useful to disentangle 5f versus 6d contributions and to test whether the ligand-induced trends that we observed here for La hold for Ac.**

Response to the reviewer's comments

We thank the Reviewer for their positive assessments and for acknowledging our efforts in addressing all the comments.

Reviewer 1

Recommendation — I greatly appreciate the authors' efforts in addressing all of my comments. The manuscript has been significantly improved and is now ready for publication. I would like to congratulate the authors once again on this excellent work.

Response: Once again, many thanks for the positive evaluation and for all the constructive comments, which definitely helped improve the quality of our manuscript.